



# Measurement report: Variations and environmental impacts
# of atmospheric $N_2O_5$ concentrations in urban Beijing during
# the 2022 Winter Olympics
Tiantian Zhang[1], Peng Zuo[2], Yi Chen[2], Tong Liu[1], Linghan Zeng[2], Weili Lin[1,3*], Chunxiang
Ye [2*]
[1]NEAC Key Laboratory of Ecology and Environment in Minority Areas, College of Life and Environmental
Sciences, Minzu University of China, Beijing, 100081, China
[2]State Key Joint Laboratory for Environmental Simulation and Pollution Control, College of Environmental
Sciences and Engineering, Peking University, Beijing, 100871, China
[3]Institute of National Security, Minzu University of China, Beijing, 100081, China
*Correspondence to*: Weili Lin (linwl@muc.edu.cn) and Chunxiang Ye (c.ye@pku.edu.cn)



**Abstract.** The chemistry of nitrate radical ($NO_3$) and dinitrogen pentoxide ($N_2O_5$) plays a pivotal role in
tropospheric nighttime chemistry. Given their close linkage to precursor variations, emission reduction during
the 2022 Beijing Winter Olympics likely affected $NO_3$ and $N_2O_5$ behavior. In this study, we measured $N_2O_5$,
$NO_2$, $O_3$, etc. during and after the Olympics, and compared pollutant levels as well as the contributions of
reaction pathways to the loss of $NO_3$ and $N_2O_5$. Throughout the entire observation period, $NO_3$ production rate
averaged $0.5 \pm 0.4$ ppbv $h^{-1}$, and the $N_2O_5$ mixing ratio could reach up to 875 pptv within 1 min, indicating their
active production. The relatively long $\tau(N_2O_5)$ at night, with an average of $11.9 \pm 11.8$ minutes, suggested a
slow rate of $N_2O_5$ loss during the winter season. Despite low NO (below 3 ppbv), it dominated $NO_3$ loss (79.0 %).
VOCs oxidation contributed 0.2 %, mainly from styrene. During the Olympics, emission reductions led to
decreased NO and VOCs, which in turn reduced their reaction with $NO_3$. The heterogeneous uptake of $N_2O_5$,
another $NO_3$ loss pathway, accounted for 20.8 % during the event and 10.6 % afterward. This uptake is crucial
for $NO_3$ removal at night, and would be essential for winter nitrate formation in urban Beijing.

## 1 Introduction

Nitrate radical ($NO_3$) and dinitrogen pentoxide ($N_2O_5$) play crucial roles in the nocturnal atmospheric chemical
cycle, controlling the removal and conversion of nitrogen oxides ($NO_x$) and volatile organic compounds (VOCs).
They significantly contribute to the formation of nitrate and secondary organic aerosols during the nighttime
(Crutzen, 1979; Wayne et al., 1991). $NO_3$ primarily originates from the reaction between $NO_2$ and $O_3$,
encapsulated in the reaction scheme (R1). Given the swift thermodynamic equilibrium (R2) between $NO_3$ and
$N_2O_5$, these species are frequently considered in tandem. During daytime, the rapid photolysis of $NO_3$ (R3) and
its reaction with NO (R4) result in a short $NO_3$ lifetime (< 5 s). Consequently, the concentrations of $NO_3$ and
$N_2O_5$ usually fall below the detection limit of analytical instruments during daylight hours.
The direct removal pathways of $NO_3$ include heterogeneous reactions on the surface of particulate matter and
gas-phase reactions with NO (R4) and VOCs (R5), which can influence the atmospheric lifetimes of nighttime
$NO_x$ and VOCs (Ng et al., 2017; Wayne et al., 1991). $NO_3$ is also capable of reacting with alkenes in an addition
reaction (R5) and subsequently with $O_2$ to form $RO_2$, which further generates organic nitric acid, one of the
important sources of secondary organic aerosols (Fry and Sackinger, 2012; Hoyle et al., 2007; Pye et al., 2010).
The removal pathways of $N_2O_5$ represent indirect removal pathways for $NO_3$ chemistry (R6), primarily
involving reactions with water vapor and heterogeneous reactions on cloud droplets and particle surfaces
(Brown and Stutz, 2012; Chang et al., 2011).
$NO_2 + O_3 \rightarrow NO_3 + O_2$ (R1)
$NO_2 + NO_3 \rightleftarrows N_2O_5$ (R2)
$NO_3 + h\nu \rightarrow O_2 + NO$ (R3)



$NO_3 + NO \rightarrow 2NO_2$                                                                              (R4)
$NO_3 + VOC \rightarrow$ products                                                                          (R5)
$N_2O_5 + H_2O \ (l) \rightarrow 2HNO_3$                                                                    (R6)
In recent years, anthropogenic emission control measures have played a pivotal role in improving air pollution
in China (Li et al., 2020; Zhang et al., 2016). Despite a declining trend in $NO_x$ emissions in China over the past
decades, the emission intensity remains relatively high (Li et al., 2024). Reactive nitrogen-containing
compounds have emerged as a prominent factor in China's complex air pollution scenario (Zhu et al., 2023).
With advancements in measurement techniques, several research teams have explored the core processes of
reactive nitrogen species in atmospheric pollution, particularly in regions with severe atmospheric pollution
such as the North China Plain, the Yangtze River Delta, and the Pearl River Delta (Li et al., 2020; Tham et al.,
2016; Wang et al., 2018, 2020, 2024, 2013; Yun et al., 2018). While $NO_3$ reactivity is typically attenuated under
low-temperature winter conditions, thereby restricting its oxidative capacity, multiple studies—including winter
campaigns such as Yun et al. (2018) and Yan et al. (2023)—have demonstrated the significance of nocturnal
$NO_3$ chemistry in cold seasons.
Recent studies have indicated that the increase in the nocturnal $NO_3$ production rates leads to enhanced
nocturnal oxidation (Wang et al., 2023a). Under the influence of emission reduction policies, nocturnal $NO_3$
radicals in urban areas may experience an "explosive" increase (Wang et al., 2023b), and the contribution of
nocturnal nitrogen chemistry to winter haze formation in the Beijing area has risen (Yan et al., 2023). Despite
these efforts, there is still a lack of comprehensive understanding and continuous monitoring of the chemical
processes of reactive nitrogen-containing compounds like $NO_3$ and $N_2O_5$.
During the 2022 Winter Olympics, a series of emission reduction measures were implemented in and around
Beijing to safeguard air quality. As a result, the average $PM_{2.5}$ concentration in Beijing was 24 μg m$^{-3}$, a
significant improvement compared to historic levels during the same period (Huang et al., 2024). Given their
close linkage to precursor variations, the 2022 Beijing Winter Olympics' emission cuts likely affected $NO_3$ and
$N_2O_5$ behavior. In this study, we carried out observational research during and after the Olympics to understand
the nocturnal chemistry of $NO_3$ and $N_2O_5$ in urban Beijing, to unravel the balance and loss mechanisms of $NO_3$
and $N_2O_5$, and to assess the impacts of emission reduction measures on their nocturnal chemistry.
**2 Methods**
**2.1 Site description**
Measurements were conducted from 5 February to 3 March 2022. The study site was situated on the rooftop of
the NO. 1 Science Building at Peking University in Beijing (39.99° N, 116.31° E, 61.6 m asl). As shown in Fig.



1, the location is proximal to two traffic arteries along the North Fourth Ring Road, rendering it representative
of an urban site, as corroborated by previous studies (Hu et al., 2023; Wang et al., 2017b; Yao et al., 2023).

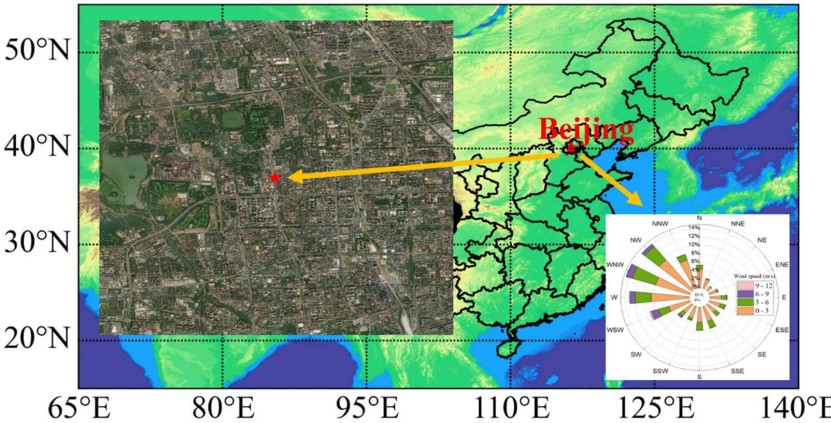


**Figure 1. Measurement Site, Surroundings & Wind Rose (Winter 2022).**

The sampling inlets were positioned at a height of 1.5 meters above the rooftop, approximately 20 meters above
ground level. Throughout the measurement period, the prevailing wind direction was predominantly from the
northwest, with an average wind speed of $2.0 \pm 2.0$ m s$^{-1}$.

**2.2 Instrument setup**

In this study, the ambient concentrations of $NO_3$ were determined utilizing an in-house-developed cavity ring-
down spectroscopy (CRDS) analyzer, whereas $N_2O_5$ was quantified through its thermal decomposition to $NO_3$.
A comprehensive elucidation of the measurement techniques and parameter quantification methodologies is
elaborated upon in our prior research (Zhang et al., 2024). Due to a malfunction in the $NO_3$ measurement
chamber, only the aggregate concentration of $N_2O_5 + NO_3$ could be ascertained. The instrument's limit of
detection was established at 2.9 pptv ($1\sigma$), with an associated measurement uncertainty of $\pm 13.7$ %. For the
calibration of the observational data, a stable source of $N_2O_5$ and $NO_3$ generation was employed to quantify the
losses incurred during gas path transport, encompassing both tubing and filter losses (Zhang et al., 2026). It
revealed that the tubing loss amounted to 11.4 %, while the filter membrane loss was recorded at $4.5 \% \pm 0.5$ %.
Other measured parameters include NO, $NO_2$, $O_3$, VOCs, and meteorological factors. NO and $O_3$ mixing ratios
were measured using commercial instruments, specifically the Model 42i-Y and Model 49i from Thermo Fisher
Scientific (USA). $NO_2$ mixing ratios were observed via a cavity-enhanced absorption spectroscopy (CEAS)
(Zhou et al., 2022). VOC concentrations were determined using a gas chromatograph equipped with mass
spectrometry and flame ionization detectors (Wang et al., 2014). Meteorological parameters, including wind
direction, wind speed, temperature (T), and relative humidity (RH), were monitored utilizing a sensor





meteorological measurement system (Metone, USA). The PM$_{2.5}$ concentration data were obtained from the
Beijing Municipal Ecological and Environmental Monitoring Center (bjmemc.com.cn). Detailed information
about these instruments is listed in Table 1.

**Table 1 Species measured in the field work**

| Species | Time resolution (s) | Limit of Detection/working range | Methods | Accuracy (%) | References |
|---|---|---|---|---|---|
| O$_3$ | 60 | 1 ppbv (parts per billion by volume) | UV photometry | ± 1 % | - |
| NO$_2$ | 30 | 8 pptv | CEAS | < 6 % | (Zhou et al., 2022) |
| NO | 60 | 50 pptv | chemiluminescence | ± 1 % | - |
| N$_2$O$_5$ | 60 | 2.9 pptv | CRDS | ± 18 % | (Zhang et al., 2024) |
| VOCs | 3600 | 1–26 pptv | GC-MS/FID | 0.8 % – 6.1 % | (Wang et al., 2014) |
| T | 60 | -50–50℃ | A three-element composite thermistor | ± 0.1℃ | www.metone.com |
| RH | 60 | 0–100 % | Thin film polymer capacitor | ± 0.2 % | www.metone.com |

**2.3 Calculation methods**
When N$_2$O$_5$ concentrations are obtained, the concentration of NO$_3$ can be determined by dividing the
concentration of N$_2$O$_5$ by the equilibrium constant $K_{eq}$ and the concentration of NO$_2$ (Osthoff et al., 2006; Wang
et al., 2017c), which is specified in Eq. (1).
$$[NO_3]=\frac{[N_2O_5]}{K_{eq}[NO_2]}$$ (1)
Here, $K_{eq}$ represents the temperature-dependent equilibrium constant established when NO$_3$ attains steady-
state equilibrium with N$_2$O$_5$, and is given by $5.50\times10^{-27}\times \exp(10724/T)$, where $T$ is the temperature in Kelvin
(Wang et al., 2024).
The primary source of NO$_3$ and N$_2$O$_5$ is the chemical reaction of NO$_2$ with O$_3$. Consequently, the concentrations
of NO$_2$ and O$_3$ are key factors influencing the production rate of NO$_3$ ($P$(NO$_3$)). This production rate is
mathematically represented by Eq. (2) (Brown et al., 2011). Assuming that the formation and loss processes of
NO$_3$ and N$_2$O$_5$ are in a state of dynamic equilibrium, the lifetime of N$_2$O$_5$, denoted as $\tau_{N_2O_5}$, can be expressed
as the ratio of its concentration to the rate of NO$_3$ production, as determined by Eq. (3) (Brown and Stutz, 2012;
Lin et al., 2022; Wang et al., 2017b).



$P(\text{NO}_3) = k_{\text{NO}_2+\text{O}_3} \times [\text{NO}_2] \times [\text{O}_3]$                                                                           (2)
$\tau_{\text{N}_2\text{O}_5} = \dfrac{[\text{N}_2\text{O}_5]}{P(\text{NO}_3)} = \dfrac{[\text{N}_2\text{O}_5]}{k_{\text{NO}_2+\text{O}_3}\cdot[\text{NO}_2]\cdot[\text{O}_3]}$                                                         (3)
The primary sinks for $\text{NO}_3$ can be attributed to the following four mechanisms:

(a) Photolytic decomposition,

(b) Reaction with NO,

(c) Reaction with VOCs,

(d) Heterogeneous uptake by $\text{N}_2\text{O}_5$.

The total $\text{NO}_3$ reactivity ($k_{\text{NO}_3}$) is determined using Eq. (4) (Wang et al., 2020). The nocturnal $\text{NO}_3$ loss rate,
denoted as L($\text{NO}_3$), is calculated via Eq. (5).
$k_{\text{NO}_3} = j(\text{NO}_3) + k_{\text{NO}_3+\text{NO}}\cdot[\text{NO}] + k_{\text{NO}_3+\text{VOCs}}\cdot[\text{VOCs}] + k_{\text{N}_2\text{O}_5}\cdot K_{\text{eq}}[\text{NO}_2]$                        (4)
L($\text{NO}_3$) = $\sum k_i[\text{VOC}_i]\cdot[\text{NO}_3] + k_{\text{NO}_3+\text{NO}}\cdot[\text{NO}][\text{NO}_3] + k_{\text{N}_2\text{O}_5}\cdot[\text{N}_2\text{O}_5]$                           (5)
Here, $j(\text{NO}_3)$ denotes the photolysis rate constant for $\text{NO}_3$ decomposition. The rate coefficients $k_{\text{NO}_2+\text{O}_3}$ and
$k_{\text{NO}_3+\text{NO}}$ correspond to the rate coefficients for reaction Eqs. (1) and (4), respectively, as referenced in Atkinson
et al. (2004). The reactivity of $\text{NO}_3$ with VOCs is characterized by a first order loss rate constant, calculated as
the product of the reaction rate constant $k_i$ and the VOC concentrations [$\text{VOC}_i$].
$k_{\text{N}_2\text{O}_5}$ represents the total first-order loss rate coefficient for the heterogeneous uptake of $\text{N}_2\text{O}_5$ at the aerosol
surface, which is governed by the uptake coefficient $\gamma(\text{N}_2\text{O}_5)$, the aerosol surface area density $S_a$ ($\mu\text{m}^2\,\text{cm}^{-3}$),
and the mean molecular speed of $\text{N}_2\text{O}_5$, c. $\gamma(\text{N}_2\text{O}_5)$ is influenced by the components of particulate matter and
atmospheric environmental conditions (e.g., humidity and temperature) (Bertram et al., 2009; Tang et al., 2014;
Yu et al., 2020).
$k_{\text{N}_2\text{O}_5} = \frac{1}{4} c S_a \gamma(\text{N}_2\text{O}_5)$                                                                                (6)
Under the steady-state assumption, $\gamma(\text{N}_2\text{O}_5)$ is determined by the slope of the linear regression between
$K_{\text{eq}}[\text{NO}_2]\tau(\text{N}_2\text{O}_5)^{-1}$ and $\frac{1}{4}cS_a\gamma(\text{N}_2\text{O}_5)K_{\text{eq}}[\text{NO}_2]$, as formalized in Eq. (7) and extensively applied to field
observations (Brown et al., 2016; Lin et al., 2022; Lu et al., 2022). $S_a$ is calculated from the $\text{PM}_{2.5}$ concentration
using the empirical formula presented in Eq. (8) (Zhang et al., 2022).
$K_{\text{eq}}[\text{NO}_2]\,\tau(\text{N}_2\text{O}_5)^{-1} = \frac{1}{4}cS_a\gamma(\text{N}_2\text{O}_5)\,K_{\text{eq}}[\text{NO}_2] + k_{\text{NO}_3}$                                     (7)
$S_a = 60.03 \times [\text{PM}_{2.5}]^{0.62}$                                                                     (8)




## 3 Results

### 3.1 Measurements overview

Figure 2 illustrates the time-series variations in the mixing ratios of $N_2O_5$ and associated trace gases, alongside meteorological parameters, captured during the 2022 Beijing Winter Olympics (BWO) at a temporal resolution of 1 minute. Valid data were systematically acquired over a 26-day span, from 5 February to 3 March. In alignment with the 2022 Beijing Winter Olympics timeline, the observation interval was bifurcated into two distinct periods: (1) the Olympic Games Period (OGP; spanning 5-20 February), and (2) the Post-Olympics Period (POP; extending from 21 February to 3 March). Comprehensive statistical metrics for each period are meticulously detailed in Table 2.

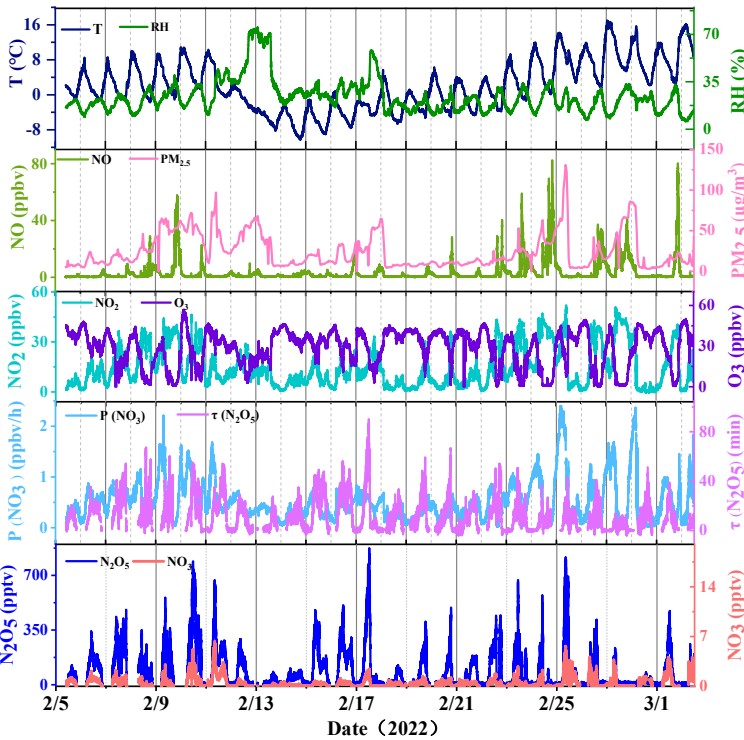

**Figure 2. Time series for $N_2O_5$, $NO_3$, related trace gases, and meteorological data (T and RH) measured at Beijing from 5 February to 3 March 2022.**

During the OGP, nocturnal temperatures predominantly lingered below the freezing point. A heavy snowfall event, accompanied by elevated relative humidity, occurred between February 13 and February 14, coinciding with a $PM_{2.5}$ concentration of 68 $\mu g\ m^{-3}$. Transitioning into the POP, ambient temperatures exhibited a marked



escalation from -0.4 °C to 5.6 °C, coupled with a decline in relative humidity. This period was punctuated by
an episode of heightened $PM_{2.5}$ pollution, with concentrations peaking at 131 μg m⁻³. The average $PM_{2.5}$
concentration throughout the entire observational period was $24 \pm 21$ μg m⁻³ and the average value of $S_a$ was
$402 \pm 215$ μm² cm⁻³.
**Table 2. Summary of observed parameters for the two periods (mean ± standard deviation).**

| Species | OGP | | POP | |
|---|---|---|---|---|
| | All day | Nighttime | All day | Nighttime |
| $O_3$ (ppbv) | 29.9 ± 9.5 | 27.4 ± 10.3 | 26.7 ± 10.6 | 19.8 ± 12.1 |
| $NO_2$ (ppbv) | 12.6 ± 8.2 | 14.5 ± 9.3 | 18.2 ± 12.3 | 20.7 ± 13.1 |
| NO (ppbv) | 1.9 ± 2.3 | 1.0 ± 1.2 | 5.7 ± 6.1 | 4.8 ± 6.0 |
| $N_2O_5$ (pptv) | 87.3 ± 71.6 | 137.6 ± 112.7 | 62.1 ± 57.7 | 97.8 ± 90.3 |
| $NO_3$ (pptv) | 0.4 ± 0.4 | 0.6 ± 0.6 | 0.3 ± 0.4 | 0.5 ± 0.6 |
| $PM_{2.5}$ (μg m⁻³) | 25 ± 2 | 26 ± 2 | 23 ± 3 | 23 ± 2 |
| T (°C) | -0.4 ± 3.9 | -1.4 ± 3.6 | 5.6 ± 3.9 | 3.5 ± 3.5 |
| RH (%) | 27.1 ± 13.3 | 28.6 ± 12.8 | 19.3 ± 4.3 | 20.3 ± 3.8 |
| $P(NO_3)$ (ppbv h⁻¹) | 0.5 ± 0.2 | 0.5 ± 0.2 | 0.6 ± 0.4 | 0.5 ± 0.3 |
| $\tau(N_2O_5)$ (min) | 10.9 ± 17.0 | 17.0 ± 17.0 | 7.4 ± 4.4 | 11.6 ± 6.8 |

During the observation period, the mean concentration of $O_3$ was $28.6 \pm 12.8$ ppbv, notably lower than spring
values observed in Beijing (Wang et al., 2018). Nocturnal $O_3$ concentrations were much higher during the OGP
than those during the POP, suggesting enhanced $NO_3$ production rates under comparable $NO_2$ levels during the
OGP.
The average $NO_x$ concentration was $18.2 \pm 16.6$ ppbv, substantially lower than autumn values typically observed
at this site (generally exceeding 30 ppbv) (Li et al., 2022; Wang et al., 2017b). During morning rush hours, the
peak NO mixing ratio reached 66.6 ppbv, with an average of $3.5 \pm 7.2$ ppbv, a value lower than that reported
for Beijing in September ($6.1 \pm 14.5$ ppbv) (Wang et al., 2017b), and significantly below the $50.2 \pm 51.4$ ppbv
observed by Li (Li et al., 2022).
In contrast to the POP, nocturnal NO mixing ratios decreased during the OGP, suggesting that the
implementation of emission reduction policies effectively curbed primary pollutant emissions during the Winter
Olympics. The subsequent increase in NO emissions facilitated its reaction with $O_3$ to form $NO_2$, leading to
higher $NO_2$ mixing ratios during the POP compared to OGP. Elevated NO levels rapidly consumed $NO_3$, leading
to swift decreases in both $NO_3$ and $N_2O_5$ concentrations.
During the observation period, the $N_2O_5$ concentrations exhibited notable fluctuations, with a mean daily value
of $86.7 \pm 116.5$ pptv. As shown in Fig. 2, the maximum $N_2O_5$ mixing ratio occurred on February 18 at 00:15,
reaching a value of 874.9 pptv. This value, while lower than the values previously reported for the Huairou site



in Beijing during the winter and the same site during the autumn (both greater than 1 ppbv) (Li et al., 2018;
Wang et al., 2017b), surpassed the maximum value observed during the spring at the same site (Wang et al.,
2017a). At the time of the peak $N_2O_5$ concentration, the measured mixing ratios of $NO_2$ and $O_3$ were 14.6 ppbv
and 26.8 ppbv, respectively, with NO levels remaining relatively low at just 0.4 ppbv. The mean $NO_3$ mixing
ratios, derived from $N_2O_5$ thermal equilibrium calculations, were $0.6 \pm 0.7$ pptv, markedly lower than
observational values reported for Shanghai ($16 \pm 9$ pptv) (Wang et al., 2013).
Table 3 presents a comparative analysis of the mixing ratios of $N_2O_5$ and $NO_3$, $P(NO_3)$, and $\tau(N_2O_5)$ across
various regions of China. During the Winter Olympics observation period, the concentrations of $NO_3$ precursors
($NO_2$ and $O_3$) at this site were lower than those observed in rural areas, such as Wangdu in Hebei province
(Wang et al., 2022). This discrepancy resulted in relatively lower calculated $P(NO_3)$ values, which ranged from
0.01 to 2.4 ppbv h$^{-1}$, with a mean value of $0.5 \pm 0.4$ ppbv h$^{-1}$. The average $P(NO_3)$ value observed in this study
aligns with the data reported for summer at the Mount Tai site ($0.45 \pm 0.40$ ppbv h$^{-1}$) and winter in the Beijing
area (0.4 ppbv h$^{-1}$) (Wang et al., 2021, 2017d). However, these values are considerably lower than those
observed in Beijing during autumn and in Taizhou during summer (Li et al., 2020; Wang et al., 2017b). In
contrast to national statistics, the average $P(NO_3)$ value in this study is lower than the cold season (October-
March) average for China, which stands at $0.67 \pm 0.23$ ppbv h$^{-1}$ (Wang et al., 2023a).
**Table 3. Summary of field-observed $N_2O_5$, $NO_3$, $NO_2$, and $O_3$ mixing ratios, $P(NO_3)$ and $\tau(N_2O_5)$.**

| Region | Location | Time | $N_2O_5$ (pptv) | $NO_3$ (pptv) | $NO_2$ (ppbv) | $O_3$ (ppbv) | $P(NO_3)$ (ppbv h$^{-1}$) | $\tau(N_2O_5)$ (min) | References |
|---|---|---|---|---|---|---|---|---|---|
| Urban | Beijing | Sep-Oct, 2016 | $68.0 \pm 136.7$ | - | $35.1 \pm 16.6$ | $27.7 \pm 25.2$ | $2.25 \pm 2.02$ | - | (Wang et al., 2017b) |
| Urban | Beijing | Nov-Dec, 2016 | $23.4 \pm 25.9$ (max 43.0) | $0.04 \pm 0.09$ | $39.0 \pm 17.8$ | $16.6 \pm 8.1$ | $0.94 \pm 0.83$ | $2.5 \pm 2.5$ | (Li et al., 2022) |
| Urban | Beijing | May-June, 2016 | 73 (max 937) | 8 (max 133) | 14.4 | 40.8 | $1.2 \pm 0.9$ (night) | $270 \pm 240$ s | (Wang et al., 2018) |
| Urban | Shanghai | Aug-Oct, 2011 | $310 \pm 380$ | $16 \pm 9$ (max 95) | ~76 | $23 \pm 8$ (max 57) | $1.10 \pm 1.09$ | - | (Wang et al., 2013) |
| Urban | Changzhou | May-June, 2019 | $53.4 \pm 66.1$ (max 304.7) | $4.7 \pm 3.5$ (max 17.7) | $13.7 \pm 8.9$ | $48.4 \pm 27.8$ | $1.7 \pm 1.2$ (night) | $1.6 \pm 1.5$ | (Lin et al., 2022) |
| Suburban | Taizhou | May-Jun, 2018 | $26.0 \pm 35.7$ (max 492) | $4.4 \pm 2.2$ (max 29.3) | $14.0 \pm 10.0$ | $48.2 \pm 32.5$ | $1.2 \pm 0.3$ | $0.93 \pm 1.13$ | (Li et al., 2020) |
| Rural | Wangdu | Jun- July, 2014 | $30.5 \pm 35.4$ (max 429) | $4.8 \pm 3.3$ (max 25) | 9.6 | 54 | $1.03 \pm 0.48$ | 162 s | (Wang et al., 2022) |
| Mountain | Shandong | Jul- Aug, 2014 | $6.8 \pm 7.7$ (max 167) | - | 3 (night) | 77 (night) | $0.45 \pm 0.40$ | 1.2–1.3 (night) | (Wang et al., 2017d) |



| Region | Location | Time | $N_2O_5$ (pptv) | $NO_3$ (pptv) | $NO_2$ (ppbv) | $O_3$ (ppbv) | $P(NO_3)$ (ppbv h$^{-1}$) | $\tau(N_2O_5)$ (min) | References |
|--------|----------|------|-----------------|---------------|---------------|--------------|---------------------------|----------------------|------------|
| Urban | Beijing | Feb-Mar, 2022 | $86.7 \pm 116.5$ | $0.6 \pm 0.7$ (max 4.6) | $14.8 \pm 11.5$ | $28.6 \pm 12.8$ | $0.5 \pm 0.4$ | $11.9 \pm 11.8$ | This study |

The relatively lower $P(NO_3)$ values in this study can be attributed to the lower temperatures, which reduce the
reaction rate constants for $NO_2$ and $O_3$. For instance, under identical $NO_2$ and $O_3$ mixing ratios ($NO_2$=15 ppbv
and $O_3$=30 ppbv), an increase in temperature from -1 °C to 5 °C leads to a corresponding increase in the reaction
rate constant from $1.59 \times 10^{-17}$ to $1.94 \times 10^{-17}$, resulting in a rise in $P(NO_3)$ from 0.70 ppbv h$^{-1}$ to 0.83 ppbv h$^{-1}$.
The temporal trend of $P(NO_3)$ during the POP exhibited a slight difference from that during the OGP, which
can be attributed to fluctuations in temperature and precursor concentrations.
$\tau(N_2O_5)$ serves as an important parameter for delineating the sources and sinks of $NO_x$ within a given region. It
plays a crucial role in elucidating the chemical equilibrium dynamics of $NO_3$ and $N_2O_5$ at measurement sites
and in evaluating the capacity for $N_2O_5$ removal. In this study, the observed $\tau(N_2O_5)$ is notably prolonged
compared to those documented in Wangdu (77 s and 172 s) and Beijing (270 s) (Tham et al., 2016; Wang et al.,
2018), yet they remain shorter than those of the measurements conducted in the Hong Kong boundary layer
(Brown et al., 2016). This discrepancy underscores a diminished reactivity of $NO_3$-$N_2O_5$ at our study site during
the observation window, leading to a more gradual overall depletion of $NO_3$ and $N_2O_5$.
$\tau(N_2O_5)$ varied significantly between the two periods, with nocturnal mean lifetime during the POP being
approximately 5 minutes shorter than that recorded during the OGP. This observation points towards an
intensified nocturnal depletion of $NO_3$ and $N_2O_5$ in the urban setting of Beijing, a phenomenon that will be
subjected to a more in-depth analysis in the forthcoming discussion sections. Drawing from Eq. (3), it is evident
that the disparities in the lifetimes mentioned above are primarily due to the variations in the concentration of
$N_2O_5$ rather than merely alterations in $P(NO_3)$. A consistent correlation was observed, wherein elevated $N_2O_5$
concentrations were associated with extended lifetimes, highlighting the pivotal role of $N_2O_5$ concentration in
modulating its own atmospheric persistence.
**3.2 Mean diurnal variations**
Figure 3 displays the average diurnal variations in NO, $NO_2$, $N_2O_5$, $NO_3$, $O_3$ mixing ratios, and the $NO_3$
production rate during the study period. Specifically, panel (a) presents the daily mean patterns for the OGP,
whereas panel (b) depicts those for the POP. Notable differences in concentration but similar diurnal trends
were observed between the two periods.





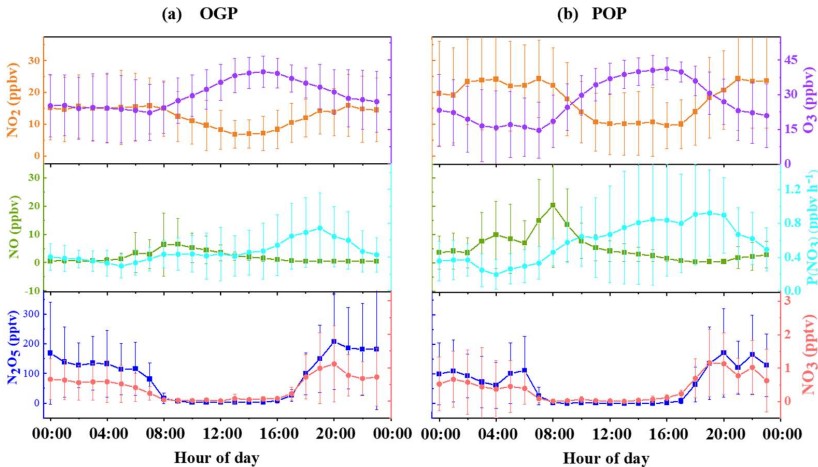

**Figure 3. Mean diurnal variations in NO, NO$_2$, N$_2$O$_5$, NO$_3$, O$_3$ mixing ratios and *P*(NO$_3$) during and after the 2022 Beijing Winter Olympics**

In urban Beijing, nocturnal NO mixing ratios were substantially lower during the OGP compared to the POP. NO concentrations exhibited a gradual increase after midnight, attaining the first peak value of 3.5 ppbv at 06:00. A secondary peak occurred during morning rush hour (08:00 LST), reaching 20.4 ppbv during the POP. O$_3$ mixing ratio exhibited a characteristic mid-afternoon peak, rising progressively after sunrise to a maximum of 41.2 ppbv between 15:00-16:00, followed by a gradual nocturnal decline to the minimum mixing ratio around 07:00. Notably, nocturnal O$_3$ levels during the POP were markedly lower than those during the OGP, indicating enhanced O$_3$ titration by elevated NO concentrations. This process contributed to maintaining lower nighttime O$_3$ levels while concurrently generating higher NO$_2$ concentrations during the POP compared to the OGP.

From the daily average variation perspective, the trends of NO$_2$ and O$_3$ mixing ratios exhibited an inverse correlation, consistent with previous studies (Wang et al., 2020). Nocturnal NO$_2$ concentrations substantially surpassed daytime levels. Furthermore, due to NO titration of NO$_3$ (R4), *P*(NO$_3$) showed a strong negative correlation with NO (Fig. S1). The highest value of *P*(NO$_3$) in both periods occurred at 19:00, with values of 0.74 ppbv h$^{-1}$ and 0.92 ppbv h$^{-1}$, respectively, both slightly lower than those observed in the North China Plain (1 ~ 2 ppbv h$^{-1}$) (Wang et al., 2022, 2018). The daily average values of *P*(NO$_3$) during the POP and the OGP were 0.5 ppbv h$^{-1}$ and 0.6 ppbv h$^{-1}$, respectively. Following the conclusion of the Winter Olympics, increased primary pollutant concentrations led to elevated *P*(NO$_3$), with the concentrations of NO$_3$ and N$_2$O$_5$ expected to rise amidst constant losses.

The diurnal variations in N$_2$O$_5$ and NO$_3$ mixing ratios exhibited comparable patterns across both periods. Specifically, both species exhibited rapid accumulation after sunset, reaching peak values around 20:00 (208.2 pptv in OGP vs. 171.2 pptv in POP). Subsequently, due to constrained NO emissions and a reduction in *P*(NO$_3$),



their concentrations gradually decreased prior to midnight, eventually reaching detection limits by sunrise.
However, during the POP, specifically from 02:00 to 04:00, elevated nocturnal NO emissions triggered a
precipitous drop in $N_2O_5$ and $NO_3$ concentrations. This decline was further exacerbated by the ensuing titration
reaction of NO, which led to a rapid decrease in $O_3$ concentration, thereby fostering higher $NO_2$ levels and an
uptick in $P(NO_3)$. Around 06:00, another $N_2O_5$ concentration peak was observed at 112.1 pptv. $NO_3$ exhibited
a similar trend, peaking at 1.1 pptv between 19:00 and 20:00. The daily average variation trends of both $N_2O_5$
and $NO_3$ aligned with those reported for the Yangtze River Delta and North China regions (Li et al., 2020; Wang
et al., 2022, 2017c; Xia et al., 2021). While the chemical conditions in this study bore similarities to those in
summer Beijing, the meteorological conditions differed, notably characterized by higher relative humidity
during the summer. The average nocturnal $N_2O_5$ concentration over the observation period was 113.7 ± 103.3
pptv, which was higher than that observed in the Changping area of Beijing (Wang et al., 2018), indicating that
the loss process of $NO_3$ and $N_2O_5$ in Beijing during winter is more sluggish compared to that in the summer.
**4 Discussion**
**4.1. The lifetime of $N_2O_5$**
RH and $S_a$ are pivotal factors influencing $\tau(N_2O_5)$ (Brown et al., 2017; Lin et al., 2022). The correlation between
these parameters and $\tau(N_2O_5)$ is presented in Fig. 4. As shown in Fig. 4(a), $\tau(N_2O_5)$ exhibited an increasing
trend with RH when RH was below 35 %. However, substantial deviations in this relationship suggests complex
underlying processes. When RH exceeded 35 %, $\tau(N_2O_5)$ decreased with increasing humidity. Specifically, at
RH > 60 %, $\tau(N_2O_5)$ approached zero, indicative of high-humidity conditions such as rain or snow. This strong
RH dependence highlights the importance of heterogeneous $N_2O_5$ uptake in urban Beijing, potentially
dominating $NO_3$ loss pathways under elevated RH. The rationale for this is twofold: (1) hygroscopic aerosols
grow at higher RH, increasing their specific surface area and thus reactive sites for $N_2O_5$ uptake; (2) this
hygroscopic growth likely enhances the efficiency of heterogeneous reactions involving $N_2O_5$, accelerating its
loss (Brown et al., 2016).
Figure 4(b) illustrates that the mean $\tau(N_2O_5)$ gradually increases from ~10 to 15 minutes as $S_a$ rises to 280 μm²
cm⁻³. However, beyond this $S_a$ value, $\tau(N_2O_5)$ remained at a relatively stable but slightly decreasing value (< 15
min) despite continued growth in $S_a$. While this declining trend in $\tau(N_2O_5)$ with increasing $S_a$ aligns with
previous findings (Lin et al., 2022; Wang et al., 2020; Zhou et al., 2018), quantitative discrepancies in the
observed values warrant note. When $S_a$ ranges from 450 to 1000 μm² cm⁻³, a robust negative correlation between
$\tau(N_2O_5)$ and $S_a$ emerged. The joint dependence of $\tau(N_2O_5)$ on RH and $S_a$ underscores the critical role of $N_2O_5$
heterogeneous uptake on particle surfaces in mediating $NO_3$ and $N_2O_5$ loss pathways. Under high-RH conditions,
this process may dominate due to enhanced hygroscopic particle growth, which increases reactive surface area
and facilitates $N_2O_5$ hydrolysis (Brown et al., 2016; Brown and Stutz, 2012; Chang et al., 2011).



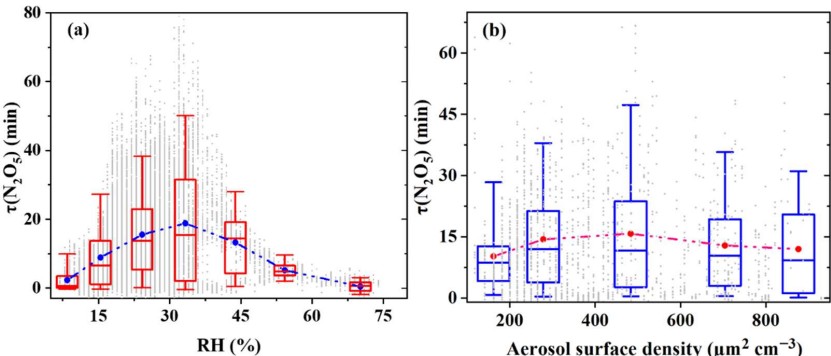


**Figure 4. The relationship between $\tau(N_2O_5)$ and $S_a$ as well as RH during the observation period.**
**4.2 NO₃ and N₂O₅ pathways**
**4.2.1 Loss of NO₃ via reaction with VOCs and N₂O₅ uptake coefficients**
To gain a deeper insight into the loss mechanisms of $NO_3$ and $N_2O_5$, the total reactivity of $NO_3$ was computed
by employing Equation (4) in Section 2.3. A comprehensive array of 99 distinct VOCs was detected at this site.
The reaction rate constants for the interaction between VOCs and the oxidizing agent $NO_3$ were obtained from
the literature (Atkinson and Arey, 2003; Brown et al., 2011) or extracted from the National Institute of Standards
and Technology database (accessible via http://webbook.nist.gov/chemistry/). For certain VOC species where
quantitative laboratory reaction rate constants were unavailable, these values were estimated based on the
reaction rate constants of analogous species.
A detailed examination of the observational data disclosed that the VOCs with the highest concentrations were
predominantly anthropogenic VOCs (AVOCs), including ethane, propane, acetone, acetylene, and ethylene.
Despite their elevated concentrations, their low reaction rate constants with $NO_3$ resulted in negligible
contributions to $k_{NO_3}$, thereby indicating that they were not primary reactants. The observational data for
Biogenic VOCs (BVOCs) solely comprised isoprene, with an average concentration of 0.016 ppbv.
Nevertheless, owing to its comparatively high reaction rate with $NO_3$, isoprene emerged as a significant
contributor to the $NO_3$ loss pathway associated with VOCs. The statistical data pertaining to VOC
concentrations and reaction rates are provided in Table S1.
Figure 5 presents a detailed visualization of the reactivity and relative contribution of diverse VOC categories
to $NO_3$ reactions across two periods. It revealed that AVOCs dominated nocturnal VOC-$NO_3$ reaction dynamics,
accounting for nearly 70 % of the total reactivity during both the OGP and POP. This result aligned
harmoniously with observations from other urban environments (Lin et al., 2022). The $NO_3$-oxidized VOC
landscape in Beijing was predominantly characterized by styrene and isoprene, with average reactivity values
of $0.34 \times 10^{-3}$ s⁻¹ and $0.25 \times 10^{-3}$ s⁻¹, respectively. Isoprene in urban Beijing exhibits dual anthropogenic and





biogenic sources: traffic emissions (e.g., gasoline exhaust) dominate anthropogenic contributions, while plant
emissions (e.g., deciduous and evergreen species) constitute the primary biogenic source. Remarkably, biogenic
isoprene concentrations significantly outstrip those from traffic emissions (Cheng et al., 2018; Yuan et al., 2009).
Styrene emissions in Beijing are predominantly linked to vehicle exhaust, though natural emissions from
evergreen and oleander species also contribute (Hu et al., 2023; Li et al., 2014). Other AVOCs also contribute
to $k_{NO_3}$, with an average reactivity of $0.26 \times 10^{-3}$ s$^{-1}$. When ranked by relative contribution, the dominant AVOCs
following the order: butene, butadiene, and propylene (Fig. S2). However, the reactivity estimates for BVOCs
may be underestimated, as observational data for key BVOCs (e.g., limonene, α-pinene) were unavailable in
this study.

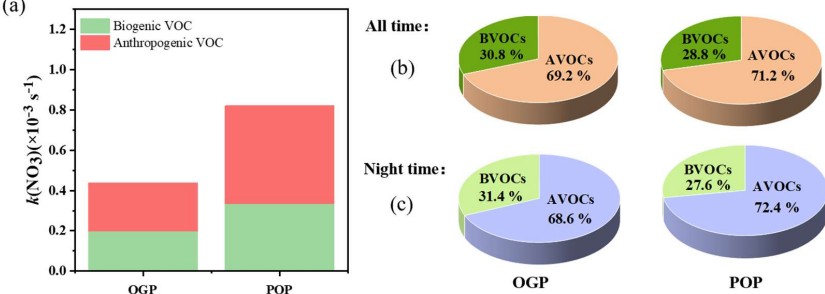


**Figure 5. The changes in reactivity and relative contribution of AVOCs and BVOCs at different periods**.
As shown in Fig. 5(a), the reactivity of VOCs with NO$_3$ underwent a marked escalation after the conclusion of
the Winter Olympics, surging from $0.44 \times 10^{-3}$ s$^{-1}$ to $0.82 \times 10^{-3}$ s$^{-1}$. The upsurge is primarily driven by elevated
VOCs concentrations following the cessation of emission reduction measures (Table S1). Statistical analyses
indicated a substantial surge in AVOCs concentrations, which accounted for 72.4 % of nocturnal reactivity
during POP. In contrast, BVOCs reactivity showed limited growth, resulting in a decreased BVOCs contribution
(28.8 % during the POP, down from 31.4 % to 27.6 %; Fig. 5c). Based on the aforementioned analysis, VOCs
concentrations remained low during the winter observation period, predominantly stemming from
transportation emissions. Stringent emission controls implemented during this period suppressed VOC
emissions, thereby reducing NO$_3$-VOC reactivity and diminishing their role in NO$_3$ radical removal. The post-
Olympic relaxation of these controls led to a rebound in AVOC levels, reasserting their dominance in nocturnal
NO$_3$ reactivity.
For the quantification of $k_{N_2O_5}$, the γ(N$_2$O$_5$) must first be determined and subsequently calculated according to
Eq. (6). The methodology for determining γ(N$_2$O$_5$) is elaborated in Section 2.3, and the calculation results for
$S_a$ are presented in Fig. S3. To mitigate the interference of water vapor and NO concentrations on the steady-
state equilibrium, data fitting was constrained to periods with RH < 70 % and NO mixing ratios < 1 ppbv. For
each day, the slope of the linear regression yielded the γ(N$_2$O$_5$), while the intercept represented $k_{NO_3}$. Due to



variations in the composition and concentration of VOCs, which can alter $k_{NO_3}$, negative intercept values
occasionally occurred during fitting, contradicting physical feasibility. Such cases were excluded from analysis,
yielding 23 valid data points. The derived $\gamma(N_2O_5)$ values were provided in Table S2 and exhibited substantial
variation under different environmental conditions. Specifically, under rainy or snowy conditions with elevated
RH, $\gamma(N_2O_5)$ would reach as high as 0.22. For most of the time, $\gamma(N_2O_5)$ fluctuated between 0.01 and 0.12, with
an average uptake coefficient of 0.032 ± 0.049. When compared to ground-based $N_2O_5$ uptake coefficients
observed in other regions of China and other countries, the values derived in this study align with those reported
in Beijing (0.01–0.09) (Li et al., 2022; Wang et al., 2017b; Zhou et al., 2018; Xia et al., 2021), but exceed those
in Wangdu (0.006–0.034) (Tham et al., 2018), Hong Kong (0.014 ± 0.007) (Brown et al., 2016), Germany
(0.028 ± 0.029), and measurements from the United States (0.008–0.04) (Mielke et al., 2013; Phillips et al.,

2016).

**4.2.2 Time-series variations in NO₃ and N₂O₅ reactivities**
The time series of $k_{NO_3}$ derived from Eq. (4) is presented in Fig. 6. By comparing reaction pathways, $NO_3$ was
found to predominantly react with NO, with an average reaction rate of 2.54 s$^{-1}$. Following this, the most notable
reaction was the heterogeneous reaction of $N_2O_5$, with an average loss rate of 0.20 ± 0.63 s$^{-1}$. During the high-
humidity night of February 13, the heterogeneous reactivity of $N_2O_5$ surged to 2.35 s$^{-1}$. A comprehensive
summary of nocturnal mean reactivities is provided in Table 4.

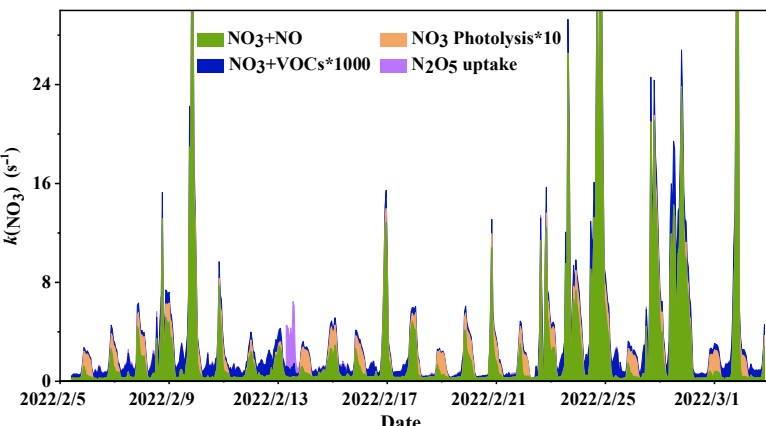


**Figure 6. Time series variation of $k_{NO_3}$ (reactions with NO and VOCs, homogeneous uptake of N₂O₅ and**
**photolysis of NO₃).**
A significant uptick in $k_{NO_3}$ was observed after the end of the Winter Olympics, escalating from 1.14 s$^{-1}$ to 3.06
s$^{-1}$. This surge suggests an augmentation in the nocturnal $NO_3$ loss, coinciding with the relaxation of emission
control measures. Concurrently, the reactivity of $NO_3$ with NO also witnessed a substantial rise, from 0.81 s$^{-1}$
to 3.00 s$^{-1}$. This enhancement can be attributed to the heightened primary NO emissions due to the cessation of



Winter Olympics-related control measures, coupled with an accelerated loss of $NO_3$ through the temperature-
dependent acceleration of the $NO_3+NO$ reaction. In a parallel trend, the reactivity of VOCs also exhibited an
increase, escalating from $0.8 \times 10^{-3}$ $s^{-1}$ to $1.4 \times 10^{-3}$ $s^{-1}$. During the POP, the $N_2O_5$ removal rate declined relative
to the OGP. This reduction is attributed to lower RH, which diminished $S_a$ availability for heterogeneous uptake
despite stable aerosol mass concentrations. Lower RH suppressed $N_2O_5$ hydrolysis, thereby reducing its
contribution to $NO_3$ loss.
**Table 4 Statistics of $k_{NO_3}$ across various pathways and time periods**

| $k_{NO_3}$ $(s^{-1})$ | OGP | POP |
|---|---|---|
| $NO_3+NO$ | 0.81 | 3.00 |
| $NO_3+VOCs$ | $0.8 \times 10^{-3}$ | $1.4 \times 10^{-3}$ |
| $N_2O_5$ uptake | 0.32 | 0.06 |
| Total | 1.14 | 3.06 |

**4.2.3 The $NO_3$ loss budget**
Figure 7 shows the diurnal variations and relative contributions of $NO_3$ loss pathways. The average $NO_3$ removal
rate increased from 1.14 ppbv $h^{-1}$ during the OGP to 1.61 ppbv $h^{-1}$ during POP, reflecting the relaxation of
emission controls. During the OGP, the $NO_3 + NO$ reaction activity exhibited a pronounced morning peak (96.7 %
of the total $k_{NO_3}$) at 07:00 local time, coinciding with morning rush-hour NO emission maxima. In contrast, the
POP featured a reconfigured diurnal pattern, with the $NO–NO_3$ reaction activity intensifying nocturnally and
peaking at 2.04 ppbv $h^{-1}$ at 22:00, accounting for 89.2 % of the $NO_3$ loss (Fig. S4).

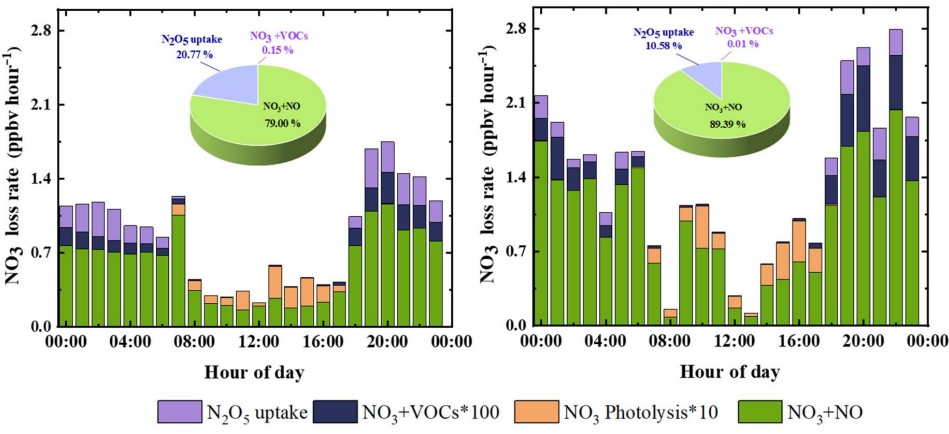


**Figure 7. Mean daily variation and reactivity share of different loss pathways.**
The heterogeneous $N_2O_5$ uptake pathway, an indirect nocturnal sink, displayed a gradual post-sunset increase
in removal rate, peaking at 0.33 ppbv $h^{-1}$ at 19:00 before declining. This pathway contributed 20.8 % of total



$NO_3$ loss during the OGP, comparable to urban Beijing (Li et al., 2022). However, its contribution diminished
to 10.6 % during the POP due to lower RH, which reduced $S_a$ availability for uptake. Similarly, the nocturnal
reaction rate between $NO_3$ and VOCs exhibited a gradual increase after sunset, reaching its peak value of
approximately 0.004 ppbv h⁻¹ after 20:00. This trend is consistent with the observation in suburban Yangtze
River Delta and urban Beijing (Hu et al., 2023; Wang et al., 2020) but is significantly lower than winter urban
sites in Beijing (0.22 ppbv h⁻¹) (Hu et al., 2023) and Changzhou (0.18 ppbv h⁻¹) (Lin et al., 2022). The maximum
VOC reactivity contribution to $NO_3$ loss was only 0.5 %, far below the 36.4 % reported for Taizhou, Jiangsu
(Wang et al., 2020), likely due to lower VOC reactivity in winter.
While $NO_3$ removal in both periods was predominantly driven by reactions with NO (Fig. 7), the relative
contributions of competing reaction pathways exhibited notable variations. During the OGP, $NO_3$ + NO
reactions accounted for 79.0 % of total $NO_3$ removal (down from 89.4 % in the POP), reflecting reduced NO
emissions under control measures. In contrast, the POP saw $NO_3$ + NO reactivity dominate (up to 96.8 % of
$NO_3$ loss), exceeding the 65 % maximum observed in urban Beijing (Li et al., 2022). The $N_2O_5$ heterogeneous
pathway's contribution (20.8 % during OGP) underscores its potential as a significant nocturnal $NO_3$ sink under
reduced NO emissions.
**5 Summary and conclusions**
This study presents a comprehensive characterization of nocturnal $NO_3$–$N_2O_5$ chemistry during the 2022
Beijing Winter Olympics. The average $N_2O_5$ concentration was 86.7 pptv, peaking at 874.9 pptv, while
maximum calculated $NO_3$ concentrations reached 4.6 pptv. The mean $P(NO_3)$ value was 0.5 ± 0.4 ppbv h⁻¹,
aligning with previous winter observations in Beijing, peaking at 2.4 ppbv h⁻¹, which reflects the region's
enhanced nocturnal oxidative capacity. The average steady-state lifetime of $N_2O_5$ was 11.9 ± 11.8 minutes,
which was longer than values reported for summer in Beijing, suggesting a slower nocturnal $NO_3$ loss rate in
winter.
NO was the dominant sink for $NO_3$ throughout the observation period, contributing up to 89.2% of its total loss,
while VOCs played a negligible role (approximately 0.1 %). Among VOCs, anthropogenic species (AVOCs)
exhibited relatively high reactivity, with styrene identified as the most reactive compound. Steady-state
calculations of the $N_2O_5$ heterogeneous uptake coefficient yielded values ranging from 0.01 to 0.1, comparable
to observations in Beijing and Wangdu. Under high relative humidity, the contribution of heterogeneous $N_2O_5$
uptake to $NO_3$ loss increased notably.
After the Winter Olympics, the lifetime of $N_2O_5$ decreased by approximately 5 minutes, indicating an enhanced
nocturnal $NO_3$ loss rate. This trend was primarily driven by an increase in the $k_{NO_3}$. During the 2022 Beijing
Winter Olympics, heterogeneous uptake of $N_2O_5$ contributed up to 20.8 % of total $NO_3$ loss, underscoring its
significance as a nocturnal $NO_3$ removal pathway under emission mitigation strategies. This finding underscores



the growing role of heterogeneous chemistry in urban NO$_3$ cycling as primary NO and VOC emissions decline.
The findings offer new insights into how emission control measures shape nighttime oxidation processes in a
polluted urban environment. Future studies incorporating detailed measurements of VOC components could
further reveal the response of regional air quality to nocturnal oxidation mechanisms.
**Data availability.** Data are available at https://doi.org/10.5281/zenodo.15381990 (T. Zhang et al., 2025).
**Supplement.** The following file is available free of charge. Supplement of "Measurement report: Variations
and environmental impacts of atmospheric N$_2$O$_5$ concentrations in urban Beijing during the 2022 Winter
Olympics"
**Author Contributions.** TZ, WL, and CY designed the research. WL, and CY organized this field campaign.
TZ, PZ, YC, TL and LZ carried out the field measurements and provided the field measurement dataset. TZ
performed data analysis, interpreted the data and wrote the manuscript with revision mainly from WL. All
authors have given approval to the final version of the manuscript.
**Competing interests.** The contact author has declared that none of the authors has any competing interests.
**Disclaimer.** Publisher's note: Copernicus Publications remains neutral with regard to jurisdictional claims in
published maps and institutional affiliations.
**Acknowledgment.** The authors would like to thank the field campaign team for the data that they contributed
during the 2022 Beijing Winter Olympics.
**Financial support.** This study was financially supported by the National Natural Science Foundation of China
(42405083), the National Key Research and Development Program (2024YFC3711902).

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
