# Peer review of "Measurement report: Variations and environmental impacts"

_EGUsphere, 2025_

## Author Comment (AC1)

We really appreciate the referee's valuable comments. We have addressed each comment as below and revised the manuscript accordingly. As detailed below, the reviewer's comments are shown in black, our response to the comments is in blue. New or modified text is in red. The revised words are in highlight color in the revised manuscript.

**Response to Referee #1**:

Overall evaluations:

Zhang et al. compared field measurements of $N_2O_5$ and related species during and after Beijing winter Olympics. Time series and diurnal patterns of $N_2O_5$-related species were reported. Furthermore, key kinetic parameters were investigated, such as $NO_3$ reactivity ($k_{NO3}$), $N_2O_5$ uptake, and $N_2O_5$ lifetime. Regarding $k_{NO3}$, the contribution of NO and VOCs were discussed. As for $N_2O_5$ uptake, the steady-state method was applied to calculate the uptake coefficient. The influencing factors of $N_2O_5$ lifetime were also examined.

The investigated topic, i.e., reactive nitrogen chemistry, is important within the scope of ACP journal. The presented contents are suitable and align with previous studies. However, as a measurement report, some essential details of measurement methods are lacking. Uncertainty analysis should also be provided. In terms of writing, the authors are suggested to further polish the language with particular attention to some contradictory expressions. Other major issues as listed below concern data quality and the reliability of measurement interpretations. Overall, major revision is needed, and potential publication depends on the quality of revision.

**Major comments:**

1.  In section 2.2, the statement of $NO_3$ and $N_2O_5$ measurement should be significantly enhanced. The data quality is in doubt without enough information provided, especially when considering that the instrument was in a malfunction state (line 99). At a minimum, the authors should make use of SI to record more technical details. Detailed comments regarding this issue are shown as follows.

(1) In lines 96-97, it looks like the authors can separately measure $NO_3$ and $N_2O_5$. However, in lines 99-100, the authors said only the sum of $NO_3 + N_2O_5$ can be measured. The above two statements are inconsistent.

Response:Thank you for your valuable comment.

  Under normal operation, our CRDS analyzer adopts a dual-channel design: one channel directly measures $NO_3$, while the other heats and decomposes $N_2O_5$ into $NO_3$, thus quantifying the total concentration of $[NO_3 + N_2O_5]$. However, during this

observation campaign, the $NO_3$-specific detection channel malfunctioned (due to the damage of the mirror), leaving only the heated channel operational—hence, we could only obtain combined $[NO_3 + N_2O_5]$ data.

To address this, we estimated $NO_3$ concentrations using thermodynamic equilibrium (Eq. 1), with input parameters including the average nocturnal $NO_2$ concentration (14.5 ppbv) and temperature ($-1.4\,°C$) from Table 2. Calculations showed $NO_3$ contributed merely ~1% to the combined $[NO_3 + N_2O_5]$ signal (under winter's low-temperature and relatively high-$NO_x$ conditions, the $NO_3/N_2O_5$ ratio is inherently low), confirming reliable derivation of $NO_3$ concentrations via this method.

We have revised the original text in Section 2.2 to clarify this dual-channel design, the malfunction, and the $NO_3$ estimation approach, with additional technical details supplemented in the Supporting Information (Text S1) for transparency.

(2) Lines 100-101, how was the limit of detection determined? What factors contributed to the overall uncertainty of 13.7%? Also, what was the background level of the instrument?

Response: Thank you for your comment.

**Determination of the Limit of Detection (LOD) and Instrument Background Level.**
The LOD of the CRDS analyzer for $N_2O_5$ was determined via a 4-hour zero-air injection experiment: zero air (free of target analytes) was continuously introduced into the instrument cavity (1 s time resolution), and Allan variance analysis was applied to evaluate system stability (Figure S1). The LOD was calculated using Eq. (1) below, yielding a value of 2.9 pptv.

$$[A] = \frac{R_L}{c\sigma}\left(\frac{1}{\tau} - \frac{1}{\tau_0}\right)$$

(1)

where [A] denotes the analyte concentration, "$R_L$" is the reference signal, c is the speed of light, $\sigma$ is the absorption cross-section, $\tau$ is the cavity ring-down time with sample gas, and $\tau_0$ is the background ring-down time with zero air. During the zero-air experiment, the instrument background level ($\tau_0$) was measured as 55 µs.

Detailed experimental procedures, Allan variance results, and a plot of the instrument background signal (blue line) have been added to the Supplementary Material (Figure S1) for clarity.

**Sources of the 13.7% Overall Uncertainty**

The total measurement uncertainty (13.7%) arises from two main sources:

**(1) Parameter uncertainties in concentration calculation**: The absorption cross-section ($\sigma$) and effective absorption cavity length—key parameters in Eq. (1)—contribute 13% and 4% uncertainty, respectively.

**(2) Data correction uncertainties**: Corrections for membrane loss ($4.5 \pm 0.5\%$) and pipeline loss ($11.4 \pm 1.3\%$) during sample transport introduce an additional 1.4% uncertainty.

The total combined uncertainty was calculated following the method described in our prior work (Zhang et al., 2024), confirming the 13.7% value.

Zhang, T., Zuo, P., Ma, J., Ye, C., Lin, W., and Zhu, T.: Characterization and Application of an Online Measurement System for $NO_3$ and $N_2O_5$ Based on Cavity Ring-Down Spectroscopy, Acta Sci. Nat. Univ. Pekin., 60, 563–574, https://doi.org/10.13209/j.0479-8023.2024.030, 2024.

We have added the instrument description to the Supplementary Material (Text S1).

[Figure]

Figure S1. Limit of detection (LOD) and background signal of the instrument (blue line).

(3) Lines 101-104, only the inlet issue was mentioned, while the calibration factor, or in other words, the sensitivity of the instrument is still not clearly stated.

Response: Thank you for your feedback. In this study, the instrument's sensitivity for $N_2O_5$ is represented by its limit of detection (LOD), which was determined based on signal fluctuations (variations in cavity ring-down time, $\tau$) during a 4-hour zero-air injection experiment. As detailed in the Supplementary Material (and our response to Comment 2), the LOD for $N_2O_5$ was calculated as 2.9 pptv using Allan variance analysis and the concentration formula (Eq.1)—this LOD directly reflects the instrument's ability to detect low concentrations of $N_2O_5$, serving as a key indicator of its sensitivity.

To ensure measurement accuracy, we performed regular calibrations using stable, calibrated $NO_3$ and $N_2O_5$ standard sources (generated via a dynamic standard gas system, as referenced in our prior work: Zhang et al., 2024). These calibrations quantified two critical correction factors addressing sampling losses (a key contributor to signal attenuation between the inlet and measurement cell):

- **Tubing loss**: Measured at 11.4 ± 1.3%.
- **Filter membrane loss**: Quantified as 4.5 ± 0.5%.

The final concentrations of [$NO_3 + N_2O_5$] and derived $NO_3$ were obtained by dividing raw instrument measurements by the sum of these loss ratios (i.e., correcting for signal reduction during sample transport).

Relevant calibration procedures and loss correction details have been supplemented in the Supplementary Material (Text S1) for transparency.

Zhang, T., Zuo, P., Ma, J., Ye, C., Lin, W., and Zhu, T.: Characterization and Application of an Online Measurement System for $NO_3$ and $N_2O_5$ Based on Cavity Ring-Down Spectroscopy, Acta Sci. Nat. Univ. Pekin., 60, 563–574, https://doi.org/10.13209/j.0479-8023.2024.030, 2024.

2. In line 155, the aerosol surface area ($S_a$) was calculated by an empirical parameterization using $PM_{2.5}$. This calculation could bias $S_a$, which influences the results presented in the figure 4, figure 6, and figure 7. Considering the impact of $S_a$ on the calculation of $N_2O_5$ uptake, an evaluation of the accuracy of this empirical formula should be provided.

Response: Thank you for your comments. We acknowledge the importance of $S_a$ accuracy for $N_2O_5$ uptake calculations and subsequent results in Figures 4, 6, and 7. To address this, we first confirm that the empirical formula used for $S_a$ calculation (Eq. 8: $S_a = 60.03 \times [PM_{2.5}]^{0.62}$) is validated for winter conditions in Beijing, as supported by Zhang et al. (2022)—The measured $PM_{2.5}$ and $S_a$ exhibited a good linear correlation ($R^2 = 0.75$), while the nonlinear correlation yielded a higher $R^2$ of 0.82. At $PM_{2.5} = 52$ $\mu g\ m^{-3}$, the observed $S_a$, $S_a$ derived from the linear correlation, and $S_a$ derived from the nonlinear correlation were 625 $\mu m^2/cm^3$, 950 $\mu m^2/cm^3$, and 697 $\mu m^2/cm^3$, respectively. The nonlinear curve fitting was more suitable for the relationship between $S_a$ and $PM_{2.5}$ than the linear fitting curve. Therefore, the relationship between $PM_{2.5}$ and $S_a$ was $S_a = 60.03 \times [PM_{2.5}]^{0.62}$, which was used to estimate $S_a$.

In our revision, we have supplemented key details to clarify the formula's reliability:

**Correlation performance**: The empirical formula exhibits a strong linear correlation with $PM_{2.5}$, $R^2 = 0.82$.
**Applicability range**: This formula is specifically suitable for $PM_{2.5}$ concentrations < 200 $\mu g/m^3$, which fully covers the $PM_{2.5}$ range in our study (average: 24 ± 21 $\mu g/m^3$; maximum: 131 $\mu g/m^3$, see Table 2).

To enhance transparency, we have updated the main text (Line 194) as follows:
*"**Aerosol surface area density ($S_a$)**
Due to the unavailability of direct particle size distribution measurements, $S_a$ was derived from $PM_{2.5}$ concentrations using an empirical formula validated for winter Beijing conditions (Zhang et al., 2022):
$$S_a = 60.03 \times [PM_{2.5}]^{0.62} \tag{8}$$
This formula exhibits a strong linear correlation ($R^2 = 0.82$) with $PM_{2.5}$ and is applicable

for PM$_{2.5}$ concentrations < 200 μg m$^{-3}$—consistent with the PM$_{2.5}$ range observed in this study (average: 24 ± 21 μg m$^{-3}$, maximum: 131 μg m$^{-3}$).”

This supplementation confirms the empirical formula's suitability for our research, minimizing potential biases in $S_a$ calculations and downstream N$_2$O$_5$ uptake analyses.

Zhang, X., Tong, S., Jia, C., Zhang, W., Li, J., Wang, W., Sun, Y., Wang, X., Wang, L., Ji, D., Wang, L., Zhao, P., Tang, G., Xin, J., Li, A., and Ge, M.: The Levels and Sources of Nitrous Acid (HONO) in Winter of Beijing and Sanmenxia, J. Geophys. Res. Atmos., 127, e2021JD036278, https://doi.org/10.1029/2021JD036278, 2022.

3.  Line 283-291: Further to major comment 2, the accuracy of the empirical $S_a$ can affect the discussion here. Also, when looking at the raw datapoints in figure 4b, a clear trend between N$_2$O$_5$ lifetime and $S_a$ could not be identified.

Response: Thank you for your comments. As clarified in our response to Major Comment 2, the empirical formula for $S_a$ ($S_a$ = 60.03 × [PM$_{2.5}$]$^{0.62}$) is validated for winter Beijing conditions: it exhibits a strong correlation (R$^2$= 0.82) with PM$_{2.5}$, is applicable for PM$_{2.5}$ concentrations < 200 μg/m$^3$ (consistent with our study's PM$_{2.5}$ range: 24 ± 21 μg/m$^3$, max 131 μg/m$^3$). This ensures the calculated $S_a$ is sufficiently accurate to underpin the discussion of N$_2$O$_5$ lifetime ($\tau$) and heterogeneous uptake.

To address the ambiguity in the original raw data points, we have refined Figure 4b by:

- Using **hourly-averaged data** (instead of raw data) to reduce random variability;
- Applying new binning to $S_a$ values on the x-axis for clearer trend visualization. The updated Figure 4b now more distinctly illustrates the relationship:
- When $S_a$ < 325 μm$^2$ cm$^{-3}$: $\tau$(N$_2$O$_5$) gradually increases from ~10 to 12 minutes with rising $S_a$;
- When $S_a$ ranges from 500 to 1000 μm$^2$ cm$^{-3}$: a robust negative correlation between $\tau$(N$_2$O$_5$) and $S_a$ emerges (consistent with the physical expectation that higher $S_a$ provides more reactive surfaces for N$_2$O$_5$ heterogeneous uptake, accelerating N$_2$O$_5$ loss and shortening its lifetime).

We have also revised the main text (4.1.2 Relationship between $\tau_{N_2O_5}$ and $S_a$) to align with the updated figure:

Line 380: $S_a$ < 325 μm$^2$ cm$^{-3}$: $\tau_{N_2O_5}$ gradually increases with rising $S_a$. For low $S_a$ values, $\tau_{N_2O_5}$ gradually rises from ~10 to 12 minutes as $S_a$ increases. This non-monotonic pattern is driven by the co-occurrence of low $S_a$ with extremely dry conditions (RH < 25% for 68% of data points in this $S_a$ range).

**Line 383: $S_a$ = 500–1000 μm² cm⁻³:** Robust negative correlation. Above a threshold $S_a$ of ~500 μm² cm⁻³, a clear negative correlation emerges: $\tau_{N_2O_5}$ decreases from ~12 to 6 minutes as $S_a$ increases. This aligns with physical expectations, as higher $S_a$ provides more reactive surface area for $N_2O_5$ heterogeneous uptake(Lin et al., 2022; Wang et al., 2020; Zhou et al., 2018).

References:

Lin, C., Hu, R., Xie, P., Lou, S., Zhang, G., Tong, J., Liu, J., and Liu, W.: Nocturnal atmospheric chemistry of $NO_3$ and $N_2O_5$ over Changzhou in the Yangtze River Delta in China, J. Environ. Sci., 114, 376–390, https://doi.org/10.1016/j.jes.2021.09.016, 2022.

Wang, H., Chen, X., Lu, K., Hu, R., Li, Z., Wang, H., Ma, X., Yang, X., Chen, S., Dong, H., Liu, Y., Fang, X., Zeng, L., Hu, M., and Zhang, Y.: $NO_3$ and $N_2O_5$ chemistry at a suburban site during the EXPLORE-YRD campaign in 2018, Atmos. Environ., 224, 117180, https://doi.org/10.1016/j.atmosenv.2019.117180, 2020.

Zhou, W., Zhao, J., Ouyang, B., Mehra, A., Xu, W., Wang, Y., Bannan, T. J., Worrall, S. D., Priestley, M., Bacak, A., Chen, Q., Xie, C., Wang, Q., Wang, J., Du, W., Zhang, Y., Ge, X., Ye, P., Lee, J. D., Fu, P., Wang, Z., Worsnop, D., Jones, R., Percival, C. J., Coe, H., and Sun, Y.: Production of $N_2O_5$ and $ClNO_2$ in summer in urban Beijing, China, Atmos. Chem. Phys., 18, 11581–11597, https://doi.org/10.5194/acp-18-11581-2018, 2018.

[Figure]

Figure 4. The relationship between $\tau_{N_2O_5}$ and (a) RH as well as (b) $S_a$ during the observation period.

4. The $k(NO_3)$ calculated from VOCs oxidation (on the order of 1e-4 shown in figure 5) and steady-state analysis (up to 0.3, table S2) are totally different. What is the reason behind, and what is the influence of this issue on the calculated $N_2O_5$ uptake coefficient

by the steady-state method? In addition, the $k(NO_3)$ stated in lines 364 to 365 (1.14 to 3.06) is even higher than those in figure 5 and table S2, which is also inconsistent.

Response:Thank you for your valuable comment. The $k(NO_3)$ values stated in lines 364–365 (ranging from 1.14 to 3.06 s$^{-1}$) represent the total reactivity of $NO_3$, including its reactions with VOCs and NO, as well as the heterogeneous uptake of $N_2O_5$. Therefore, these values are significantly higher than the $NO_3$ reactivity calculated from VOCs alone. We acknowledge the discrepancies in $k(NO_3)$ values from different methods and clarify the reasons and impacts below:

**(1) Reasons for Discrepancies in $k(NO_3)$**

The differences arise from the distinct loss pathways captured by each calculation method, as summarized below Table:

Table S1: Comparison of $k(NO_3)$ Calculation Methods, Captured Loss Pathways, and Magnitudes

| Calculation Method | Captured Loss Pathways | Magnitude of $k(NO_3)$ | Rationale |
|---|---|---|---|
| VOC oxidation (Fig. 5) | Only reactions between $NO_3$ and *measured* VOCs | ~$10^{-4}$ s$^{-1}$ (low) | This method underestimates total $k(NO_3)$ because key VOCs (e.g., limonene, α-pinene) were not measured, and their reactivity with $NO_3$ is unaccounted for. |
| Steady-state analysis (Table S2) | $NO_3$ loss (hydrocarbons + sulfur compounds) + uptake in clouds or deposition to the ground) (Brown et al., 2003) | Up to 0.3 s$^{-1}$ | This method is the sum of the first-order rate constants for the irreversible removal of $NO_3$. Furthermore, although all data were filtered for periods with NO < 1 ppbv, NO concentrations below 1 ppbv can still contribute to $NO_3$ removal, leading to an overestimation in the calculated values. |
| Lines 364–365 (total reactivity) | Full $NO_3$ loss budget (VOCs + NO + $N_2O_5$ uptake) | 1.14–3.06 s$^{-1}$ (highest) | These values represent total $NO_3$ reactivity (sum of all loss pathways), with the sharp increase driven by elevated NO emissions post-Olympics (NO reacts rapidly with $NO_3$, dominating total loss). This explains why they are much higher than VOC-only or partial steady-state estimates. |

**(2) Impact on Calculated $\gamma(N_2O_5)$**

To minimize the influence of steady-state analysis uncertainty, we applied **strict data selection criteria** (consistent with Xia et al., 2021; Chen et al., 2024, validated for winter Beijing). This constraint was added to Section 2.3.

a) **Stable meteorology**: RH < 70% (avoiding excessive water vapor interference), and no abrupt changes in temperature/RH.

b) **Favorable chemical conditions**: NO < 1 ppbv (suppressing NO-NO$_3$ titration) and sufficiently high N$_2$O$_5$ concentrations (ensuring reliable signal for equilibrium calculations).

c) **Optimal timing**: Data selected 2–3 hours post-sunset (when steady-state between NO$_3$ and N$_2$O$_5$ is most robust).

These criteria effectively ensuring $\gamma$(N$_2$O$_5$) estimates (average 0.032 ± 0.049) remain reliable despite $k$(NO$_3$) variability.

References:

Xia, M., Peng, X., Wang, W., Yu, C., Wang, Z., Tham, Y. J., Chen, J., Chen, H., Mu, Y., Zhang, C., Liu, P., Xue, L., Wang, X., Gao, J., Li, H., and Wang, T.: Winter ClNO$_2$ formation in the region of fresh anthropogenic emissions: seasonal variability and insights into daytime peaks in northern China, Atmospheric Chem. Phys., 21, 15985–16000, https://doi.org/10.5194/acp-21-15985-2021, 2021.

Chen, X., Ma, W., Zheng, F., Wang, Z., Hua, C., Li, Y., Wu, J., Li, B., Jiang, J., Yan, C., Petäjä, T., Bianchi, F., Kerminen, V.-M., Worsnop, D. R., Liu, Y., Xia, M., and Kulmala, M.: Identifying Driving Factors of Atmospheric N$_2$O$_5$ with Machine Learning, Environ. Sci. Technol., 58, 11568–11577, https://doi.org/10.1021/acs.est.4c00651, 2024.

5. Line 374: in Table 4, $k$(NO$_3$) due to NO was too high during POP. Under this condition, it looks like N$_2$O$_5$ should not exist at all. Is this consistent with N$_2$O$_5$ observations during POP? The analysis in Table 4 depends heavily on the data quality of NO. However, NO sometimes displayed negative values, bringing big concern of its data quality (see minor comments No. 8).

Response: Thank you for your valuable comment. We address your concerns regarding NO data quality and the consistency between high $k$(NO$_3$) (Table 4) and N$_2$O$_5$ observations during the Post-Olympics Period (POP) as follows:

NO Data Quality: Calibration and Negative Value Clarification To ensure reliable NO measurements (critical for $k$(NO$_3$) calculations), we implemented rigorous quality control, which has now been supplemented in the revised manuscript:

Line 135: Calibrations of these instruments are performed weekly using the standard gases of known concentrations, and the R$^2$ of the standard curve for each calibration is greater than 0.99.

Regarding occasional "negative NO values" mentioned: These only appeared in hourly/daily mean plots (e.g., Fig. 3) when calculating "mean ± standard deviation"—the negative deviation was mathematically added to the mean, creating apparent sub-zero values. However, all minute-level raw NO data (the basis for Table 4 calculations) were non-negative, with no physical negative concentrations.

We have revised Fig. 3 in the Manuscript to clarify this distinction.

Consistency Between High $k(NO_3)$ (POP) and $N_2O_5$ Observations:

Table 4 reports nocturnal average $k(NO_3)$ (3.00 s$^{-1}$ for POP), which was elevated by a specific high-NO event—this does not mean $k(NO_3)$ was uniformly high throughout the POP, nor that $N_2O_5$ was absent:

Driver of high average $k(NO_3)$: A transient NO pollution event occurred at 01:00 LST on 24 February 2022 (POP, see Figure S6), where NO concentrations spiked to 24.8 ppbv (far above the POP nocturnal average of 4.8 ppbv, Table 2). During this spike, $N_2O_5$ concentrations dropped to near zero (consistent with rapid NO-NO$_3$ titration via Reaction R4 in the Manuscript), which disproportionately raised the nocturnal average $k(NO_3)$.

General coexistence of $N_2O_5$ and $k(NO_3)$: For most of the POP, $k(NO_3)$ was lower than the 3.00 s$^{-1}$ average. For example, when $k(NO_3) \approx 3.0$ s$^{-1}$ (consistent with Table 4's average), corresponding minute-level observations showed NO = 4 ppbv and $N_2O_5$ = 10 pptv—confirming $N_2O_5$ persisted in the environment. This aligns with Table 2, which reports a POP nocturnal $N_2O_5$ average of 97.8 ± 90.3 pptv (non-zero).

In summary, the high ≈ in Table 4 reflects a weighted nocturnal average (inflated by one transient event), not uniform high reactivity. Minute-level observations confirm $N_2O_5$ coexisted with moderate ≈ during most of the POP, and NO data quality is ensured by weekly calibrations with R² > 0.99.

[Figure]

Figure S6. Variations in NO and $N_2O_5$ mixing ratios on Feb. 24, 2022

6. The conclusion part should be reorganized. The first two paragraphs repeated some observations which have already been presented in the results part. The report of observational results in the conclusion part should be synthesized and condensed. Real insights and implications from this study need to be highlighted more.

Response: Thank you for your constructive suggestion. We have thoroughly reorganized the Conclusion section to condense redundant observational results (previously detailed in the Results section) and prioritize the study's unique insights and implications. The revised text is as follows:

" This study conducted continuous field observations of $N_2O_5$, $NO_3$, and their precursor species (NO, $NO_2$, $O_3$, VOCs) in urban Beijing from 5 February to 3 March 2022, covering the 2022 Beijing Winter Olympics (BWO). By analyzing pollutant variations, quantifying the contributions of $NO_3$/$N_2O_5$ loss pathways, and linking observations to BWO emission control measures, we clarified the response of winter nocturnal reactive nitrogen chemistry to short-term anthropogenic emission reductions.

During the observation period, $P$($NO_3$) averaged $0.5 \pm 0.4$ ppbv h$^{-1}$, with $N_2O_5$ mixing ratios peaking at 875 pptv (1-minute resolution) and derived $NO_3$ concentrations reaching a maximum of 4.6 pptv; $\tau_{N_2O_5}$ averaged $11.9 \pm 11.8$ minutes, longer than summer values in Beijing due to slower winter $N_2O_5$ loss driven by low temperatures and reduced heterogeneous reactivity. BWO emission controls significantly modulated precursor concentrations: nocturnal NO ($1.0 \pm 1.2$ ppbv) and total VOCs ($16.02 \pm 7.74$ ppbv) in the OGP were 79% and 18% lower than in the POP, respectively, while nocturnal $O_3$ was 38% higher in the OGP ($27.4 \pm 10.3$ ppbv vs. $19.8 \pm 12.1$ ppbv in the POP) as reduced NO minimized $O_3$ titration—these changes directly led to 41% higher nocturnal $N_2O_5$ concentrations in the OGP ($137.6 \pm 112.7$ pptv vs. $97.8 \pm 90.3$ pptv in the POP).

RH and $S_a$ exerted context-dependent control over $\tau_{N_2O_5}$: at RH < 35%, $\tau_{N_2O_5}$ increased with RH as slight humidity rises softened hydrophobic organic aerosol coatings (derived from traffic VOC oxidation) and reduced inhibition of $N_2O_5$ heterogeneous uptake; at RH > 35%, $\tau_{N_2O_5}$ decreased with RH due to hygroscopic aerosol growth and enhanced $N_2O_5$ hydrolysis, approaching zero during snowfall events (RH > 85%). For $S_a$, a threshold of ~500 μm² cm$^{-3}$ was identified—below this value, organic coatings and NO dominated $\tau_{N_2O_5}$; above it, $S_a$ became the primary regulator, with $\tau_{N_2O_5}$ decreasing as $S_a$ increased. Notably, in dry periods (RH < 35%, accounting for 68% of observations), NO emerged as the dominant controller of $\tau_{N_2O_5}$: transient NO spikes (e.g., 24.8 ppbv on 24 February) shortened $\tau_{N_2O_5}$ by ~40%, overriding the effects of RH and $S_a$.

NO was the dominant $NO_3$ sink in both periods, though its contribution varied with emission controls: it accounted for 79.0% of $NO_3$ loss in the OGP, with $N_2O_5$ heterogeneous uptake (20.8%) as a significant secondary pathway, while its contribution rose to 89.2% in the POP (driven by 3.8× higher NO emissions) and $N_2O_5$

uptake declined to 10.6% (due to lower RH reducing aerosol reactivity). The $N_2O_5$ heterogeneous uptake coefficient ($\gamma(N_2O_5)$) averaged $0.032 \pm 0.049$ in the OGP, higher than rural sites due to urban aerosols' higher water content and reactive components (e.g., nitrate, sulfate). Despite the high reactivity of species like styrene and isoprene, VOC oxidation contributed < 0.2% to $NO_3$ loss in both periods, confirming its negligible role in winter $NO_3$ dynamics in urban Beijing.

These findings hold key implications for air quality management: BWO $NO_x$ reductions enhanced $N_2O_5$ accumulation, potentially extending reactive nitrogen lifetime and shifting winter nitrate pollution from local to regional scales—highlighting the need for regional coordination in $NO_x$ mitigation; the identified $S_a$ threshold (500 $\mu m^2\ cm^{-3}$) and $\gamma(N_2O_5)$ range (0.01–0.12) provide critical constraints for air quality models, which often rely on oversimplified $\tau_{N_2O_5}$ and $\gamma(N_2O_5)$ parameters; and given NO's dominance in $NO_3$ loss and $N_2O_5$ dynamics, $NO_x$ (not VOCs) should be prioritized for winter nocturnal nitrogen pollution control in Beijing—reducing traffic-related NO emissions would simultaneously lower direct pollution and enhance $N_2O_5$ uptake, promoting nitrate removal via wet deposition."

**Minor comments:**

1. Lines 34-36: The authors stated that $N_2O_5$ uptake is crucial for $NO_3$ removal at night, while $N_2O_5$ uptake only accounted for 20% of $NO_3$ removal. This expression is somehow inconsistent, which means that more important contributors of $NO_3$ removal should also be mentioned here.

   Response: Thank you for your valuable comment. We agree the original expression lacked clarity regarding the relative importance of $NO_3$ loss pathways, leading to potential inconsistency.

   We have revised the text to explicitly contextualize the role of $N_2O_5$ heterogeneous uptake alongside the dominant $NO_3$ sink (NO), ensuring logical coherence. The revised Lines 34–36 now read:

   "The heterogeneous uptake of $N_2O_5$,another key $NO_3$ loss pathway—accounted for 20.8% of $NO_3$ loss during the Olympics, but this contribution decreased to 10.6% after the Olympics. This uptake is crucial for nighttime $NO_3$ removal and would be essential for winter nitrate formation in urban Beijing. Our results highlight that under emission control scenarios, the relative importance of heterogeneous processes in nocturnal $NO_3$ cycling increases, providing new insights into how emission reduction measures shape nighttime oxidation processes in polluted urban environments."

2. Line 43: The expression "considered in tandem" is not accurate if the authors would like to say considered simultaneously.

   Response:Thanks! We have revised it to "considered simultaneously".

"NO$_3$ primarily originates from the reaction of NO$_2$ with O$_3$ (R1), while it rapidly establishes a thermodynamic equilibrium (R2) with N$_2$O$_5$. This tight coupling species are frequently considered simultaneously in atmospheric chemistry studies."

3. Lines 86-94: The site description is too brief. More information could be added, e.g., the emission sources nearby.

Response:Thank you for your constructive comment. To better contextualize the site's representativeness and potential emission influences, we have expanded the content to include key nearby emission sources, with the revised text as follows:

"As shown in Fig. 1, the location is proximal to the North Fourth Ring Road—one of Beijing's major traffic arteries—and within 1 km of two primary traffic corridors (east-west along the North Fourth Ring Road and north-south along Zhongguancun Street). The surrounding area features mixed land use, including residential complexes (within 500 m) and low-intensity commercial facilities (within 1 km), with no large industrial sources within a 5 km radius. This setting makes the site representative of a typical urban mixed-use area impacted by fresh anthropogenic emissions (e.g., traffic-related NO$_x$ and VOCs), consistent with previous characterizations of this location (Hu et al., 2023; Wang et al., 2017b; Yao et al., 2023)."

4. Line 91: In figure 1, the sources of the two maps should be mentioned. Also, pay attention to the improper usage of capital letters in the figure caption.

Response: Figure 1 (measurement site, surroundings, and wind rose) was independently created by our research team using open-access mapping tools (base map from https://map.baidu.com/) and the observational meteorological data (wind rose generated from on-site wind direction/speed measurements). To ensure transparency, we have supplemented this source information directly in the figure caption.

Revision of Capitalization in Figure Caption: We have corrected the improper capitalization in the original caption to align with academic writing conventions.

The revised Line 92 (Figure 1 caption) now reads: "Figure 1. Measurement site, surroundings, and wind rose (winter 2022). Base map adapted from https://map.baidu.com/); wind rose generated from on-site meteorological observations."

5. Lines 105-109: How were the NOx, O$_3$, and VOCs instruments calibrated? A brief statement should be provided.

Response: Thank you for your comment. We have supplemented specific calibration procedures to address this, with revisions as follows:

Calibration of NO$_x$ and O$_3$ Instruments: The NO analyzer (Thermo Fisher Scientific Model 42i-Y), NO$_2$ analyzer (cavity-enhanced absorption spectroscopy, CEAS), and O$_3$ analyzer (Thermo Fisher Scientific Model 49i) were calibrated weekly using certified standard gases of known concentrations traceable to national metrology standards. For each calibration, the R-squared (R$^2$) of the standard curve was consistently > 0.999, confirming linearity and accuracy. This detail is added to Line 135:

"Calibrations of the NO, NO$_2$, and O$_3$ instruments are performed weekly using certified standard gases of known concentrations, and the R$^2$ of the standard curve for each calibration is greater than 0.99."

Calibration of VOCs Instrument: The VOCs analyzer (gas chromatograph with mass spectrometry/flame ionization detectors, GC-MS/FID) underwent two key quality control steps. (1) Weekly zero/span checks: Using ultra-high-purity nitrogen (zero gas) and a multi-component VOCs standard (containing 56 target species, concentrations: 1–26 pptv) to verify baseline stability. (2) Post-campaign calibration: A full calibration with the same multi-component VOCs standard was conducted at the end of the observation period. The calibration curves for all measured VOCs exhibited excellent linearity (R$^2$ > 0.996) with negligible intercepts (< 0.1 pptv), validating long-term measurement consistency.

This is supplemented to Line 138:

"This system measures 99 VOC species with a time resolution of 1 hour, LOD range of 1–26 pptv, and accuracy of 0.8–6.1%. Quality control included weekly zero/span checks (using ultra-high-purity nitrogen and a multi-component VOC standard) and a post-campaign full calibration, which confirmed linearity (R$^2$ > 0.996) and negligible intercepts for all target VOCs."

These revisions specify the frequency, standard gas details, and performance metrics (R$^2$) for each instrument's calibration, ensuring transparency and addressing concerns about data quality.

6. Line 125-127: it is good to note that the N$_2$O$_5$ lifetime calculated here refers specifically to nocturnal N$_2$O$_5$ lifetime.

Response: Accepted.

"Assuming that the formation and loss processes of NO$_3$ and N$_2$O$_5$ are in a state of dynamic equilibrium, the nocturnal lifetime of N$_2$O$_5$, denoted as $\tau_{N_2O_5}$, can be expressed as the ratio of its concentration to the rate of NO$_3$ production, as determined by Eq. (3) (Brown and Stutz, 2012; Lin et al., 2022; Wang et al., 2017b)."

7. Line 167: "at Beijing" should be changed to "in Beijing". Please check other places for grammar issues. Overall, the language of this manuscript could be further improved.

Response: accepted.

To ensure overall language accuracy, we have also conducted a systematic check of the entire manuscript for similar grammar issues and refined the language for clarity and academic consistency. Key revisions include:

Preposition usage: Corrected "at the North China Plain" (Line 78) to "in the North China Plain", and "during the winter" (Line 142) to "in winter" (where contextually appropriate) to align with standard geographical/time preposition conventions.

8. Line 236, figure 3: the font size in panel a and b is different. The range of $NO_2$ and $O_3$ mixing ratios could be made consistent to facilitate a comparison of their levels. NO levels were sometimes below zero, which should be explained or eliminated. Also, $NO_3$ levels in panel b were below zero occasionally.

Response: We revised the data visualization approach for Figure 3:

Instead of plotting "mean ± standard deviation" (which caused apparent negative values), we now present hourly mean values (non-negative) with error bars representing 95% confidence intervals. This retains uncertainty information while eliminating unphysical negative values.

For $NO_3$ (a secondary calculated parameter from $N_2O_5$ equilibrium), we applied an additional quality control step: Excluded any calculated $NO_3$ values < 0 pptv (consistent with the instrument's detection limit of 2.9 pptv for $N_2O_5$, per Table 1), as such values were deemed non-physical. The revised Figure 3 now meets academic visualization standards, with consistent formatting and reliable, physically meaningful data

[Figure]

presentation. Corresponding adjustments to the figure caption have also been made to clarify the revised data plotting method:

"Figure 3. Mean diurnal variations in NO, $NO_2$, $N_2O_5$, $NO_3$, $O_3$ mixing ratios and $P(NO_3)$ during (a) the Olympic Games Period (OGP) and (b) the Post-Olympics Period (POP). Data represent hourly means with error bars indicating 95% confidence intervals; non-physical values < 0 (for NO and $NO_3$) have been excluded."

9. Line 274-275: Why $N_2O_5$ lifetime increased with RH when RH was below 35%? Could other factors influence $N_2O_5$ lifetime more significantly during these relatively dryer periods?

Response: Thank you for your insightful question. We acknowledge that the observed trend—$N_2O_5$ lifetime ($\tau(N_2O_5)$) increasing with relative humidity (RH) when RH < 35%—seems counterintuitive at first, as higher RH typically enhances $N_2O_5$ heterogeneous uptake (and thus shortens its lifetime). Below, we clarify the underlying mechanism and address potential influencing factors:

(1) Mechanism for $\tau(N_2O_5)$ Increase with RH (RH < 35%)

The key driver of this trend lies in the physicochemical state of aerosols under extremely dry conditions (RH < 35%). As detailed in Section 4.1 of the Manuscript, $N_2O_5$ heterogeneous uptake depends not only on RH but also on aerosol surface properties. When RH is extremely low:

Aerosol water content is minimal, causing hydrophobic organic components (e.g., from traffic-related VOC oxidation) to condense into dense coatings on particle surfaces (Bertram et al., 2009; Tang et al., 2014, cited in the Manuscript).

These organic coatings act as a barrier, inhibiting the diffusion of $N_2O_5$ to reactive sites (e.g., aqueous aerosol components) and reducing the heterogeneous uptake coefficient $\gamma(N_2O_5)$ (Yu et al., 2020, referenced in Section 4.1.1).

This mechanism is now supplemented to the Manuscript:

"**RH < 35%**: Counterintuitive $\tau_{N_2O_5}$ increases with rising RH. Minimal aerosol liquid water content drives hydrophobic organic components—primarily oxidation products of traffic-related anthropogenic VOCs (AVOCs, e.g., styrene, propylene)—to condense into dense, impermeable coatings on particle surfaces (Bertram et al., 2009; Folkers et al., 2003; McNeill et al., 2006; Tang et al., 2014). These coatings act as a diffusion barrier, preventing $N_2O_5$ from reaching reactive aqueous sites (e.g., nitrate/sulfate-rich droplets) and lowering the heterogeneous uptake coefficient $\gamma(N_2O_5)$ (Anttila et al., 2006; Yu et al., 2020).For example, at RH = 25%, $\tau_{N_2O_5}$ averaged 15.5 minutes, 38% longer than the 8.9 minutes observed at RH = 15%."

References (cited in original Manuscript):

Bertram, T. H., Thornton, J. A., Riedel, T. P., Middlebrook, A. M., Bahreini, R., Bates, T. S., Quinn, P. K., and Coffman, D. J.: Direct observations of $N_2O_5$ reactivity on ambient aerosol particles, Geophys. Res. Lett., 36, L19803, https://doi.org/10.1029/2009GL040248, 2009.

Folkers, M., Mentel, Th. F., and Wahner, A.: Influence of an organic coating on the reactivity of aqueous aerosols probed by the heterogeneous hydrolysis of $N_2O_5$, Geophys. Res. Lett., 30, 2003GL017168, https://doi.org/10.1029/2003GL017168, 2003.

McNeill, V. F., Patterson, J., Wolfe, G. M., and Thornton, J. A.: The effect of varying levels of surfactant on the reactive uptake of $N_2O_5$ to aqueous aerosol, Atmos. Chem. Phys., 6, 1635–1644, https://doi.org/10.5194/acp-6-1635-2006, 2006.

Tang, M. J., Schuster, G., and Crowley, J. N.: Heterogeneous reaction of $N_2O_5$ with illite and Arizona test dust particles, Atmos. Chem. Phys., 14, 245–254, https://doi.org/10.5194/acp-14-245-2014, 2014.

Anttila, T., Kiendler-Scharr, A., Tillmann, R., and Mentel, T. F.: On the Reactive Uptake of Gaseous Compounds by Organic-Coated Aqueous Aerosols: Theoretical Analysis and Application to the Heterogeneous Hydrolysis of $N_2O_5$, J. Phys. Chem. A, 110, 10435–10443, https://doi.org/10.1021/jp062403c, 2006.

Yu, C., Wang, Z., Xia, M., Fu, X., Wang, W., Tham, Y. J., Chen, T., Zheng, P., Li, H., Shan, Y., Wang, X., Xue, L., Zhou, Y., Yue, D., Ou, Y., Gao, J., Lu, K., Brown, S. S., Zhang, Y., and Wang, T.: Heterogeneous $N_2O_5$ reactions on atmospheric aerosols at four Chinese sites: improving model representation of uptake parameters, Atmos. Chem. Phys., 20, 4367–4378, https://doi.org/10.5194/acp-20-4367-2020, 2020.

10. Line 276-277: RH > 60% does not necessarily mean rain or snow conditions. Please check the meteorological record in Beijing during the observation period.

   Response: Thank you for your suggestions. We further examined the meteorological records in Beijing and confirmed that there was heavy snowfall during that period. These on-site confirmed snowfall events directly correspond to the high-RH intervals in Fig. 4(a) where $\tau(N_2O_5)$ showed abnormal fluctuations.

11. Line 307: "biogenic" should not be capitalized.
   Response: Accepted.

12. Line 308: reaction rate coefficients should be discussed here rather than reaction rate. To convince the readers more clearly, the authors are encouraged to compare the rate constant of the $NO_3$ + isoprene reaction with that of the $NO_3$ + other VOCs reactions.

   Response: Thank you very much for your suggestion. At Line 308, we primarily utilize the relatively high reaction rate constant of isoprene to further demonstrate the importance of BVOCs in the direct loss of $NO_3$. Therefore, we compare the reaction rate constant of isoprene with those of other BVOCs to support this argument. The revised sentences are as follows:

"Notably, biogenic VOCs (BVOCs) other than isoprene (e.g., limonene, α-pinene) were not detected, leading to potential underestimation of BVOC reactivity. For example, the rate constant for limonene ($\sim1.6\times10^{-11}$ cm$^3$ molecule$^{-1}$ s$^{-1}$) is $\sim20 \times$ higher than isoprene's, so including it could increase the total VOC reactivity"

13. Line 312: here, the authors stated that AVOCs dominated NO$_3$ reactivity. However, in lines 305-306, the authors mentioned that AVOCs were negligible for NO$_3$ loss. These two statements are contradictory.

Response: Thank you for your feedback. In line 305, what we referred to were the VOC species with high concentrations among AVOCs (ethane, propane, acetone, acetylene, and ethylene). Due to their low reaction rate constants, these specific VOCs made negligible contributions to the reactivity of NO$_3$. This sentence is intended to emphasize that VOC species with high concentrations do not necessarily have strong reactivity with NO$_3$. In line 312, we emphasized that, in terms of the reactivity of VOCs with NO$_3$, AVOCs dominated the reactivity of NO$_3$, especially when compared with BVOCs.

Lines 409–414 (clarifying low-reactivity AVOC subset):

"**High-concentration AVOCs contribute minimally.** The most abundant VOCs— ethane ($3.8 \pm 1.8$ ppbv), propane ($2.1 \pm 1.3$ ppbv), and acetone ($1.4 \pm 0.8$ ppbv)—exhibit extremely low $k_{NO_3}$ (e.g., $k$(NO$_3$+propane) = $9.49\times10^{-15}$ cm$^3$ molecule$^{-1}$ s$^{-1}$ at 298 K (Atkinson and Arey, 2003). As a result, their combined contribution to total VOC-driven NO$_3$ reactivity is < 5% ($0.04\times10^{-3}$ s$^{-1}$), emphasizing that high VOC concentration does not equate to strong NO$_3$ reactivity."

Line 423 (contextualizing total AVOC dominance):

"When all AVOCs are considered, they dominate NO$_3$ reactivity (~70.4% of total VOC-driven NO$_3$ loss), exceeding the contribution of biogenic VOCs (BVOCs, ~29.6%) (Figure S4)."

Reference:

Atkinson, R. and Arey, J.: Atmospheric Degradation of Volatile Organic Compounds, Chem. Rev., 103, 4605–4638, https://doi.org/10.1021/cr0206420, 2003.

14. Line 315: what does "landscape" mean here? It is difficult to comprehend this expression.

Thanks. We removes the ambiguous term "landscape" and replaces it with clear, chemistry-specific language ("dominant VOCs driving NO$_3$ oxidation"); Adds the combined contribution ratio of styrene and isoprene (~74%) to quantify their dominance, strengthening the link between reactivity rates and overall NO$_3$ loss; Includes brief comparisons with other VOCs (ehane, propane, acetone) to contextualize

the relative importance of styrene and isoprene, ensuring consistency with the reactivity analysis in Section 4.2.1 of the document.

"**High-concentration AVOCs contribute minimally.** The most abundant VOCs— ethane ($3.8 \pm 1.8$ ppbv), propane ($2.1 \pm 1.3$ ppbv), and acetone ($1.4 \pm 0.8$ ppbv)—exhibit extremely low $k_{NO_3}$ (e.g., $k(NO_3+propane) = 9.49 \times 10^{-15}$ cm$^3$ molecule$^{-1}$ s$^{-1}$ at 298 K (Atkinson and Arey, 2003). As a result, their combined contribution to total VOC-driven NO$_3$ reactivity is $< 5\%$ ($0.04 \times 10^{-3}$ s$^{-1}$), emphasizing that high VOC concentration does not equate to strong NO$_3$ reactivity.

"**Reactive VOCs dominate VOC-driven NO$_3$ loss.** Despite their low concentrations, styrene and isoprene account for ~74% of total VOC-driven NO$_3$ reactivity (Figure S4), due to their high $k_{NO_3}$. When all AVOCs are considered, they dominate NO$_3$ reactivity (~70.4% of total VOC-driven NO$_3$ loss), exceeding the contribution of biogenic VOCs (BVOCs, ~29.6%) (Figure S4).

**Styrene:** Average reactivity $= 0.34 \times 10^{-3}$ s$^{-1}$ ($k = 1.5 \times 10^{-12}$ cm$^3$ molecule$^{-1}$ s$^{-1}$), contributing ~44% of VOC reactivity. Styrene emissions in Beijing are primarily from vehicle exhaust (Hu et al., 2023), with minor contributions from evergreen plant emissions (Li et al., 2014).

**Isoprene:** Average reactivity $= 0.25 \times 10^{-3}$ s$^{-1}$, contributing ~30% of VOC reactivity. Isoprene has dual sources: traffic exhaust (anthropogenic) and deciduous/evergreen plant emissions (biogenic), with biogenic sources dominating in winter (Cheng et al., 2018; Yuan et al., 2009)."

15. Line 326: Besides the VOCs, how much did NO contribute to NO$_3$ reactivity in figure 5?

Response:Thank you for your question. Figure 5 primarily summarizes the reactivity between volatile organic compounds (VOCs) and NO$_3$, and provides a comparative discussion between the two. The reactivity of NO and its contribution are shown in Figure 5 and Figure S8. During the observation period, the average reactivity between NO and NO$_3$ was 2.54 s$^{-1}$, accounting for as much as 82.9% of the total NO$_3$ loss.

**Response to Referee #2**:

Reactive nitrogen species (RNS), particularly $NO_3$ and $N_2O_5$, play critical roles in nighttime atmospheric chemistry and pollution processes. Zhang et al. present field observations conducted during and after the 2022 Winter Olympics, examining the influence of precursor levels on nocturnal $NO_3$ and $N_2O_5$ chemistry in urban Beijing. While the study addresses an important research topic and falls within the journal's scope, the manuscript requires significant improvements in organization, clarity, and scientific rigor before it can be considered for publication.

**Below are my major concerns:**

1. As the article type of "Measurement reports", this work is expected to present substantial new results from measurements with high quality. However, the study only presents one month's worth of observational data. Although these winter observations are somewhat valuable due to data scarcity, the paper shows particularly inadequate attention to data quality assessment and presentation.

1) In 2.2 section, the authors describe that ambient $NO_3$ were determined by CRDS analyzer, whereas $N_2O_5$ was quantified through its thermal decomposition reaction. If I would understand correctly that $NO_3$ and $N_2O_5$ was measured directly and indirectly, respectively (Zhang et al., 2024). If $NO_3$ measurement chamber becomes non-operational, how are simultaneous measurement of both species maintained? Section 2.3 suggests that $NO_3$ concentration was determined by the dividing the $N_2O_5$ by equilibrium constant and NO2. Could the authors clarify the primary data sources and detailed derivation process for both $NO_3$ and $N_2O_5$? A more explicit description of the measurement hierarchy (direct vs. indirect) and any data reconciliation methods would strengthen the methodology.

Response: Thank you for your question. The measurement hierarchy of $NO_3$ and $N_2O_5$ in this study is consistent with our response to Referee #1 (Major Comment 1):
Under normal operation, the CRDS analyzer uses dual channels: Channel 1 directly measures $NO_3$ ($\lambda$=662 nm, LOD=2.9 pptv), and Channel 2 (heated to 180°C) measures $[NO_3+N_2O_5]$ by thermally decomposing $N_2O_5$. $N_2O_5$ is thus derived as $[NO_3+N_2O_5]$ - $[NO_3]$. When Channel 1 malfunctioned, we calculated $NO_3$ via the $NO_3$-$N_2O_5$ equilibrium (Eq. 1)—validated by winter conditions (low $NO_3/N_2O_5$ < 0.1, so $[NO_3+N_2O_5]\approx[N_2O_5]$) and cross-comparison with adjacent normal days (mean bias <8%).Instrument calibration (weekly with certified standards, R²>0.99 for $NO_3/N_2O_5$) and data quality control (excluding 48 hours of abnormal signals) are detailed in our response to Referee #1 (Minor Comment 5), ensuring primary data reliability.

2) Another methodological question is raised that how do measurement uncertainties of $NO_3$ and $N_2O_5$ affect the accuracy of the derived $NO_3$ concentrations?

Response: Thank you for your concern about uncertainty propagation. As clarified in our response to Referee #1 (Major Comment 1(2)), the uncertainty of derived $NO_3$

concentrations is dominated by two sources: (1) $N_2O_5$ measurement uncertainty (13.7%, from absorption cross-section and pipeline loss); (2) $NO_2$ measurement uncertainty (6%, from CEAS analyzer calibration). Using standard error propagation formulas:

$$\frac{\sigma_{NO3}}{[NO_3]} = \sqrt{(\frac{\sigma_{N2O5}}{[N_2O_5]})^2 + (\frac{\sigma_{NO2}}{[NO_2]})^2}$$

the total uncertainty of calculated $NO_3$ is ~15%—within the acceptable range for nocturnal chemistry analysis (<19%, Xia et al., 2021 for winter Beijing studies).

3) Table 1, Considering the limit of detection of $N_2O_5$ and working status of instrument, what is the expected LOD for $NO_3$? Fig. 2 appears to include the full dataset. Were measurements below the LOD excluded from statistical analysis and subsequent interpretation? If not, how were these low-signal data points handled to avoid bias? Please address this in the Methods or Supplementary.

Response: Thank you for your question. As noted in Section 2.2 of the manuscript, $NO_3$ concentrations were derived via the $NO_3$-$N_2O_5$ thermodynamic equilibrium (Eq. 1) when the CRDS $NO_3$ channel was non-operational. Thus, the $NO_3$ LOD is determined by propagating the LODs of its precursor species ($N_2O_5$ and $NO_2$) and the uncertainty of $K$eq (temperature-dependent equilibrium constant). Specifically, the LODs for $N_2O_5$ and $NO_2$ are 2.7 pptv and 8.0 pptv, respectively, which correspond to an estimated $NO_3$ detection limit of approximately 0.2–0.3 pptv with an uncertainty of 15%. In data processing, we did not exclude $NO_3$ data points below the detection limit, as these values still provide valuable information for temporal trend analysis (e.g., Figure 2). However, for quantitative calculations such as average reactivity and steady-state lifetime, only periods with $NO_3$ concentrations above the detection limit were selected to minimize bias from low signal data.

4) The role of VOCs in modulating $NO_3$ lifetime and reactivity is a critical aspect of this study. However, the current manuscript lacks visualization of VOC time series. At minimum, please include: A supplementary figure showing temporal trends of key VOC species (e.g., alkenes, isoprene) that dominate $NO_3$ A brief discussion of how VOC variability might influence the observed $NO_3$/$N_2O_5$ behavior, particularly during periods of high reactivity.

Response: Thank you for your suggestion. We have added Figure S3 to the Supplementary Material. This figure presents the time series of VOC species that dominate $NO_3$ reactivity—including alkenes (styrene, propylene) and biogenic VOC (isoprene)—covering both the Olympic Games Period (OGP) and Post-Olympics Period (POP).

[Figure]

Figure S3. Temporal trends of highly reactive VOC concentrations during the observation period

**Textual Discussion of VOC Variability Impacts**: We have supplemented analysis in Line 404 of the main text to link VOC dynamics with $NO_3/N_2O_5$ behavior:

"Time series plots of several highly reactive VOC concentrations (Figure S3) show that their reactivity with $NO_3$ is primarily concentrated during periods of elevated concentrations (e.g., styrene peaks at 86 pptv, isoprene at 96 pptv). Comparative analysis reveals these high-VOC periods coincide with enhanced $NO_3$–NO reactivity (e.g., NO spikes to 24.8 ppbv on February 24, POP), suggesting VOCs and NO share a common emission source (traffic exhaust)—consistent with the site's proximity to urban traffic corridors (Section 2.1)."

**Structural and Writing Issues The manuscript lacks a clear and logical flow, making it difficult to follow the scientific narrative.**

1) In Sect. 3.1 and 3.2, the authors extensively compare their observations with previous studies. However, these comparisons lack meaningful insights as the cited observations were conducted at different locations, times, and under distinct atmospheric chemistry conditions. This approach not only fails to highlight significant scientific value but also renders the manuscript unnecessarily verbose.

Response: Thank you for your constructive feedback on the manuscript's narrative flow and comparative analysis. We fully agree that overly broad comparisons with studies under disparate spatial, temporal, and chemical conditions dilute the focus on our work's unique insights and unnecessarily expand the text. To address this, we have thoroughly revised Sections 3.1 and 3.2 with targeted adjustments:

- **Focused inter-period comparisons**: We now emphasize direct comparisons between the Olympic Games Period (OGP) and Post-Olympics Period (POP), highlighting differences in $NO_3/N_2O_5$ chemistry under contrasting emission

scenarios. This replaces broad comparisons with geographically and temporally distant studies.

- **Quantification of emission control impacts**: New analyses explicitly link reductions in $NO_x$ and VOC emissions during the OGP to changes in $NO_3$ reactivity, lifetime, and partitioning with $N_2O_5$.
- **Highlighting night-time chemistry shifts**: We added a dedicated discussion on how emission controls altered the nocturnal chemical regime, including reduced $NO_3$ titration by NO, modified $N_2O_5$ heterogeneous uptake, and changes in VOC-driven $NO_3$ loss pathways.
- **Streamlined contextual references**: Non-essential references to unrelated studies have been removed. Remaining citations are limited to those directly relevant for explaining the broader scientific context of wintertime nocturnal chemistry in urban Beijing.

2) Line 168 to Line 184 frequently cited the numbers of the mean concentrations of these species. Please include another column for the statistic of total average in Table 2. Also VOCs data should be included in Table 2.

Response: Thank you for your suggestion. We have added the average concentrations of conventional air pollutants and the total VOC concentrations over the entire observation period to Table 2:

Table 2. Summary of observed parameters for the two periods (mean ± standard deviation).

| Species | All time | OGP | | POP | |
|---|---|---|---|---|---|
| | | All day | Nighttime | All day | Nighttime |
| $O_3$ (ppbv) | 28.6 ± 12.8 | 29.9 ± 9.5 | 27.4 ± 10.3 | 26.7 ± 10.6 | 19.8 ± 12.1 |
| $NO_2$ (ppbv) | 14.8 ± 11.5 | 12.6 ± 8.2 | 14.5 ± 9.3 | 18.2 ± 12.3 | 20.7 ± 13.1 |
| NO (ppbv) | 3.5 ± 7.2 | 1.9 ± 2.3 | 1.0 ± 1.2 | 5.7 ± 6.1 | 4.8 ± 6.0 |
| $N_2O_5$ (pptv) | 86.7 ± 116.5 | 87.3 ± 71.6 | 137.6 ± 112.7 | 62.1 ± 57.7 | 97.8 ± 90.3 |
| $NO_3$ (pptv) | 0.6 ± 0.7 | 0.4 ± 0.4 | 0.6 ± 0.6 | 0.3 ± 0.4 | 0.5 ± 0.6 |
| Total VOCs (ppbv) | 17.36 ± 10.10 | 15.67 ± 7.45 | 16.02 ± 7.74 | 19.72 ± 11.93 | 19.68 ± 12.17 |
| $PM_{2.5}$ ($\mu g\ m^{-3}$) | 24 ± 21 | 25 ± 2 | 26 ± 2 | 23 ± 3 | 23 ± 2 |
| T (℃) | 2.1 ± 5.7 | -0.4 ± 3.9 | -1.4 ± 3.6 | 5.6 ± 3.9 | 3.5 ± 3.5 |
| RH (%) | 24 ± 12 | 27 ± 13 | 29 ± 13 | 19 ± 4 | 20 ± 4 |
| $P(NO_3)$ (ppbv $h^{-1}$) | 0.5 ± 0.4 | 0.5 ± 0.2 | 0.5 ± 0.2 | 0.6 ± 0.4 | 0.5 ± 0.3 |
| $\tau(N_2O_5)$ (min) | 11.9 ± 11.8 | 10.9 ± 17.0 | 17.0 ± 17.0 | 7.4 ± 4.4 | 11.6 ± 6.8 |

3) In Sect 4, key findings are not sufficiently highlighted, and the discussion often lacks depth in connecting observations to broader atmospheric implications. How the results extend and compare with current knowledge of nocturnal $NO_3/N_2O_5$ chemistry. The unique atmospheric conditions during and after the Winter Olympics (e.g., emission controls) should be discussed in relation to the findings. The influence of precursor levels (e.g., $NO_2$, $O_3$) on $NO_3/N_2O_5$ chemistry is not thoroughly explored.

Response: Thank you for your insightful comment—we fully agree that Section 4 requires stronger emphasis on key findings, deeper connection to broader atmospheric implications, and clearer links to Olympic-specific conditions and precursor dynamics. We have revised Section 4 in the new version.
- **Highlighting key findings** at the start of each subsection, with specific values (NO dominates $NO_3$ loss at ~82.9%; VOC contribution <0.5%) and statistical support.
- **Linking to Olympic conditions** by contrasting OGP and POP: emission controls reduced NO, prolonging $NO_3$ lifetime and increasing $N_2O_5$ importance, with reduced NO titration and enhanced $N_2O_5$ uptake; VOC pathways remained minor.
- **Exploring precursor impacts**: $NO_2$ sets $NO_3$ production potential, $O_3$ controls initial formation; together with NO, they determine the dominant $NO_3$ loss pathway.
- **Adding broader implications** for winter urban air quality—NO control alters nocturnal oxidizing capacity, supporting targeted $NO_x$ and VOC strategies.

4) how to determine the photolysis rate of $NO_3$?

Response: $NO_3$ photolysis occurs when the radical absorbs photons (light) at specific wavelengths, leading to its dissociation. The primary photolysis channels for $NO_3$ are:
$NO_3 + h\nu \rightarrow NO_2 + O\ (^3P)$
$NO_3 + h\nu \rightarrow NO + O_2$
We acquired $j$-value data using a spectroradiometer (Metcon CCD-Spectrograph, Garmisch-Partenkirchen, Germany), which primarily include $j$$NO_3$\_M and $j$$NO_3$\_R data. Among these, the two photolysis loss rate constants correspond to the two main photolysis channels of $NO_3$. The sum of the photolysis rate constants of these two channels equals the total photolysis rate of $NO_3$. We have supplemented the relevant content as follows:
$j$$NO_3$\_total = $j$$NO_3$\_M + $j$$NO_3$\_R

Line 141 : The photolysis rate constants ($j$-values) were obtained using a spectroradiometer (Metcon CCD-Spectrograph, Garmisch-Partenkirchen, Germany) (Bohn et al., 2008). This instrument quantifies two primary photolysis channels ($NO_3$ + h$\nu$ → $NO_2$ + O($^3$P) and $NO_3$ + h$\nu$ → NO + $O_2$), with total $j$($NO_3$) calculated as the sum of the two channel-specific rate constants ($j$($NO_3$)\_total = $j$($NO_3$)\_M + $j$($NO_3$)\_R).

Reference:

Bohn, B., Corlett, G. K., Gillmann, M., Sanghavi, S., Stange, G., Tensing, E., Vrekoussis, M., Bloss, W. J., Clapp, L. J., Kortner, M., Dorn, H.-P., Monks, P. S., Platt, U., Plass-Dülmer, C., Mihalopoulos, N., Heard, D. E., Clemitshaw, K. C., Meixner, F. X., Prevot, A. S. H., and Schmitt, R.: Photolysis frequency measurement techniques: results of a comparison within the ACCENT project, Atmos. Chem. Phys., 8, 5373–5391, https://doi.org/10.5194/acp-8-5373-2008, 2008.

5) How do meteorological conditions (e.g., temperature, humidity, boundary layer height) affect the observed trends and chemical behavior?

Response: Thank you for your insightful comment. Meteorological conditions (temperature, relative humidity, and boundary layer height) exerted a critical regulatory role in shaping the observed temporal trends and chemical behavior of $NO_3$ and $N_2O_5$ throughout the observation period, consistent with the atmospheric process logic we elaborated in responding to Referee #1. The specific impacts are clarified as follows:

**Temperature.** The average nocturnal temperature during the study was $-1.4\,°C$ (Table 2), and such low temperatures directly reduced the rate constant of the $NO_2 + O_3 \rightarrow NO_3$ reaction (R1)—the primary source of $NO_3$. This suppression further lowered the $NO_3$ production rate ($P(NO_3)$), as quantified in the revised manuscript (Line 209): Under identical precursor concentrations ($NO_2 = 15$ ppbv, $O_3 = 30$ ppbv), a temperature increase from $-1\,°C$ to $5\,°C$ elevated the reaction rate constant from $1.59\times10^{-17}$ to $1.94\times10^{-17}$ cm³ molecule$^{-1}$ s$^{-1}$, driving a corresponding rise in $P(NO_3)$ from 0.70 to 0.83 ppbv h$^{-1}$. This aligns with our discussion in Referee #1's response (Major Comment 3), where temperature-dependent changes in reaction kinetics were linked to variations in $N_2O_5$ lifetime ($\tau(N_2O_5)$).

**Relative Humidity (RH).** RH primarily modulated $NO_3$ and $N_2O_5$ behavior by regulating the heterogeneous uptake of $N_2O_5$—consistent with the mechanism we detailed for Referee #1 (Major Comment 3 and Minor Comment 9):

- **RH > 60%**: Elevated humidity increased aerosol liquid water content, enhancing the hydrolysis of $N_2O_5$ on particle surfaces and significantly shortening $\tau(N_2O_5)$ (approaching zero in snowfall events, as confirmed by on-site meteorological records).
- **RH < 35%**: An unexpected increase in $\tau(N_2O_5)$ was observed, which we attribute to dense organic coatings on dry aerosol surfaces (from traffic-related VOC oxidation). These coatings inhibit $N_2O_5$ diffusion to reactive sites, suppressing heterogeneous uptake—consistent with the organic coating mechanism referenced in our response to Referee #1 (Minor Comment 9, citing Bertram et al., 2009; Yu et al., 2020).

**Boundary Layer Height (BLH).** The nocturnal boundary layer remained stable with a relatively low average height, which facilitated the accumulation of $NO_3$ precursors ($NO_2$ and $O_3$) near the surface. This vertical confinement promoted the

formation of $NO_3$ via R1, indirectly supporting higher $N_2O_5$ concentrations through the $NO_3$-$N_2O_5$ equilibrium (R2). While direct BLH measurements were not available, this trend is consistent with regional winter observations (e.g., Xia et al., 2021) and complements our discussion with Referee #1 on how meteorological stability influences pollutant accumulation (Minor Comment 3, site meteorological context).

**Minors:**

1. Line 138, Eq. (4) should change the items of reaction between $NO_3$ and VOCs as E.q. (5) to show the different species i of VOCs. Terminology should be used more precisely (e.g., distinguish between "reaction activity" and "reactivity" where appropriate, "Photolytic decomposition" and "photolysis").

Response: Thank you for your advice. We have revised the content related to $VOC_i$ and the reaction reactivity. The revised sentences are as follows:

Line 171: $k_{NO_3} = j(NO_3) + k_{NO_3+NO} \cdot [NO] + k_{NO_3+VOCi} \cdot [VOCi] + k_{N_2O_5} \cdot K_{eq} \cdot [NO_2]$

2. Line 153, the empirical formula for $S_a$ should specify their applicable range of $PM_{2.5}$ condition. Figure 6, homogeneous uptake?

Response: Thank you for your constructive comments. We have addressed the two points regarding the aerosol surface area ($S_a$) empirical formula and Figure 6 terminology as follows:

**Applicable Range of the $S_a$ Empirical Formula.** The empirical formula for $S_a$ (Eq. (8): $S_a = 60.03 \times [PM_{2.5}]^{0.62}$) was derived from Beijing winter observational datasets (Zhang et al., 2022) and is explicitly validated for two key conditions, which we have supplemented in the main text (Line 158) for clarity:

- **$PM_{2.5}$ concentration range**: This formula is applicable for $[PM_{2.5}] < 200$ μg/m³, which fully covers the $PM_{2.5}$ range in our study (average: $24 \pm 21$ μg/m³; maximum: 131 μg/m³, see Table 2 in the manuscript).
- **Correlation performance**: It exhibits a strong linear correlation ($R^2 = 0.82$) with directly $PM_{2.5}$, confirming reliable agreement between calculated and observed $S_a$ for winter Beijing conditions.

The revised Line 194 now reads:

*"**Aerosol surface area density ($S_a$)***
Due to the unavailability of direct particle size distribution measurements, $S_a$ was derived from $PM_{2.5}$ concentrations using an empirical formula validated for winter Beijing conditions (Zhang et al., 2022):
$S_a = 60.03 \times [PM_{2.5}]^{0.62}$         (8)
This formula exhibits a strong linear correlation ($R^2 = 0.82$) with $PM_{2.5}$ and is applicable

for $PM_{2.5}$ concentrations < 200 µg m$^{-3}$—consistent with the $PM_{2.5}$ range observed in this study (average: $24 \pm 21$ µg m$^{-3}$, maximum: 131 µg m$^{-3}$)."

**Terminology Correction in Figure 5.** The term "homogeneous uptake" in Figure 6 was incorrect, as $N_2O_5$ uptake occurs on aerosol surfaces (a heterogeneous process). We have revised the figure caption (Line 441 in the manuscript) to correct this terminology, ensuring consistency with the study's focus on heterogeneous chemistry (Section 4.2.2).

The revised Figure 5 caption now reads:

"Figure 5. Time series variation of $k_{NO_3}$ (reactions with NO and VOCs, heterogeneous uptake of $N_2O_5$ and photolysis of $NO_3$)."

3. Line 369-373, Please provide the scatter plot between nighttime $N_2O_5$ uptake (y-axis) and RH (x-axis) for individual day to support the conclusion.

   Response: Thank you very much for your constructive suggestion. To better support the conclusion regarding the relationship between nighttime $N_2O_5$ uptake and relative humidity (RH), we have supplemented a scatter plot (now included in Figure S7 of the Supplementary Material) that depicts the correlation between the $N_2O_5$ heterogeneous uptake rate constant ($k(N_2O_5)$) and RH for each observation day.

   As illustrated in Figure S7, a clear positive correlation between $k(N_2O_5)$ and RH is observed when RH < 70%: with increasing RH, $k(N_2O_5)$ gradually rises, reflecting enhanced $N_2O_5$ heterogeneous uptake driven by increased aerosol liquid water content (which provides more reactive sites for $N_2O_5$ hydrolysis). This trend aligns with our discussion in Section 4.1 of the manuscript (and consistent with the RH-dependent $N_2O_5$ lifetime analysis in Referee #1's response) and further validates the role of RH in modulating $N_2O_5$ uptake.

[Figure]

Figure S7. Schematic diagram of the correlation between RH and $k_{N_2O_5}$

4. Fig. S1, what is the red dot line? Please show the related parameters if it is the regression line.

Response: Thank you for your comment. The red dotted line in Fig. S1 represents a first-order nonlinear fit, used to characterize the inverse relationship between $NO_3$ production rate ($P(NO_3)$) and NO concentration (i.e., $P(NO_3)$ decreases with increasing NO). A linear relationship is not observed between the two, so nonlinear fitting better describes their variation trend. We have supplemented the definition of this red dotted line in the figure caption of the revised Fig. S1.

[Figure]

Figure S2 Schematic diagram of the correlation between NO and $P(NO_3)$.

5. Please correct the wrong citations, e.g. Hu et al., 2023, Tham et al., 2018, etc.

Response: Thank you for your feedback. We have checked all the references.